# Lipid Nanocarriers-Enabled Delivery of Antibiotics and Antimicrobial Adjuvants to Overcome Bacterial Biofilms

**DOI:** 10.3390/pharmaceutics16030396

**Published:** 2024-03-14

**Authors:** Anam Ahsan, Nicky Thomas, Timothy J. Barnes, Santhni Subramaniam, Thou Chen Loh, Paul Joyce, Clive A. Prestidge

**Affiliations:** Centre for Pharmaceutical Innovation, University of South Australia, Adelaide, SA 5000, Australia; anam.ahsan@mymail.unisa.edu.au (A.A.); nicky.thomas@unisa.edu.au (N.T.); tim.barnes@unisa.edu.au (T.J.B.); santhni.subramaniam@unisa.edu.au (S.S.); lohty003@mymail.unisa.edu.au (T.C.L.); paul.joyce@unisa.edu.au (P.J.)

**Keywords:** lipid nanocarriers, antibiotics, antimicrobial adjuvants, biofilms, EPS-degrading enzymes, quorum sensing inhibitors, co-delivery, combination therapy

## Abstract

The opportunistic bacteria growing in biofilms play a decisive role in the pathogenesis of chronic infectious diseases. Biofilm-dwelling bacteria behave differently than planktonic bacteria and are likely to increase resistance and tolerance to antimicrobial therapeutics. Antimicrobial adjuvants have emerged as a promising strategy to combat antimicrobial resistance (AMR) and restore the efficacy of existing antibiotics. A combination of antibiotics and potential antimicrobial adjuvants, (e.g., extracellular polymeric substance (EPS)-degrading enzymes and quorum sensing inhibitors (QSI) can improve the effects of antibiotics and potentially reduce bacterial resistance). In addition, encapsulation of antimicrobials within nanoparticulate systems can improve their stability and their delivery into biofilms. Lipid nanocarriers (LNCs) have been established as having the potential to improve the efficacy of existing antibiotics in combination with antimicrobial adjuvants. Among them, liquid crystal nanoparticles (LCNPs), liposomes, solid lipid nanoparticles (SLNs), and nanostructured lipid carriers (NLCs) are promising due to their superior properties compared to traditional formulations, including their greater biocompatibility, higher drug loading capacity, drug protection from chemical or enzymatic degradation, controlled drug release, targeted delivery, ease of preparation, and scale-up feasibility. This article reviews the recent advances in developing various LNCs to co-deliver some well-studied antimicrobial adjuvants combined with antibiotics from different classes. The efficacy of various combination treatments is compared against bacterial biofilms, and synergistic therapeutics that deserve further investigation are also highlighted. This review identifies promising LNCs for the delivery of combination therapies that are in recent development. It discusses how LNC-enabled co-delivery of antibiotics and adjuvants can advance current clinical antimicrobial treatments, leading to innovative products, enabling the reuse of antibiotics, and providing opportunities for saving millions of lives from bacterial infections.

## 1. Introduction

The discovery of antibiotics was significant in human history, revolutionizing medicine and saving countless lives [1]. However, extensive use and misuse of antibiotics has resulted in the generation of multidrug-resistant (MDR) bacteria or “superbugs,” reducing their efficacy. The development of antibiotic resistance makes the treatment of persistent infections very challenging. The World Health Organization (WHO) Global Antimicrobial Surveillance System 2017 report emphasized AMR as a global challenge to health, life expectancy, and food production [2]. AMR is associated with 4.95 million deaths worldwide [3] and a global economic burden of $3.5 billion/year according to a recent modeling study by the Organisation for Economic Co-operation and Development (OECD) conducted in 2018 [4]. When comparing AMR to COVID-19, as of December 2021, there have been more than 620 million confirmed COVID-19 cases and approximately 6.5 million confirmed deaths worldwide [5]. If left unchecked, AMR could lead to around 10 million deaths per year by 2050, with economic consequences that are likely to be as severe as the 2008/2009 financial crisis [5]. Though global strategies have largely focused on discovering novel antibiotic drugs to avoid AMR, there are lesser outcomes due to limited profitability, with no new antibiotic classes having received regulatory approval since the late 1980s [6]. Further scientific and translational challenges, i.e., efflux, poor permeability, and fast resistance development, exacerbate the deficits in the antimicrobial development pipeline [7].

By existing in biofilms, a huge proportion of the bacterial population either grows slowly or exists in a dormant state, which is thought to promote tolerance [8]. Tolerance to antimicrobials is termed as the ability of a bacterial population to rapidly survive lethal antibiotic concentrations, possibly because the bacteria slow down their essential processes [9]. Generally, the activity of antibiotics largely depends on the drug’s ability to reach its target in bacterial cells at sufficient concentrations. However, a sufficient antibiotic concentration is not achieved in biofilms, as the presence of the EPS matrix delays antibiotic penetration, and in addition, there is reduced availability of bacterial targets owing to the slow growth of bacteria in biofilms [8]. Compared to tolerance, AMR is not transient, remains in the bacteria after biofilm disruption, and is instigated by mutations in the bacterial genome or attaining AMR factors via horizontal gene transfer [8]. Moreover, bacteria communicate with each other through quorum sensing, which helps to regulate their metabolism and hence the induction of biofilm formation and enhanced virulence [10].

Currently, antibiotics are the key therapeutic strategy used to treat planktonic and biofilm infections [11]. They target processes that are essential for bacterial growth and survival, including the synthesis and maintenance of cell walls and membranes or the production of DNA, RNA, or vital proteins. Unambiguously, conventional antimicrobial treatment is mostly not effective against localized and chronic infections, with biofilm-associated infections displaying up to 1000 times less susceptibility than planktonic-induced infections [7,12]. Aminoglycosides (e.g., gentamicin, amikacin, and tobramycin) are unable to penetrate the EPS owing to their molecular size and electrostatic attraction to the matrix that anchors them to the biofilm surface, resulting in reduced efficacy against biofilm-indwelling bacteria [13]. Consequently, drug concentrations within biofilms are often sub-therapeutic, which leads to a marked reduction in effectiveness, simultaneously promoting the growth of AMR. The growth of biofilms on surfaces, i.e., mucosal tissues and indwelling medical devices, as well as free-floating biofilm-like masses, is of clinical significance [14]. Although there is inadequate clinical evidence for biofilm eradication, chronic infection therapy focuses on high doses of antibiotics, often in combination with different antibiotics for long-term or surgical removal of the biofilm, where possible [15]. These approaches raise major concerns and risks of increased toxicity and complications to patients [16,17]. Hence, innovative anti-biofilm therapeutic strategies are urgently needed to overcome this serious challenge and improve clinical outcomes.

Since the development of new molecules is time-consuming and costly, combination drug treatments have been effectively used in clinical settings [18,19]. The use of non-antibiotic compounds and antibiotic adjuvants can enhance the drug’s efficacy, with the ability to target bacterial resistance when combined with other drugs. In recent years, the field of “antimicrobial adjuvants” has gained tremendous attention [20]. This review focuses on EPS-degrading enzymes and quorum sensing inhibitors (QSIs) as potential antimicrobial adjuvants targeting two major components of bacterial biofilms, i.e., the EPS matrix and QS system, representing a new era of combined antibiotic and antimicrobial adjuvant/anti-biofilm drug therapy.

Generally, the administration of antimicrobial agents in their free form has several disadvantages, e.g., a lack of site-specific delivery and the risk of the compound degrading or being cleared before reaching the target [21,22]. Furthermore, there is an increased risk of systemic toxicity when high doses of therapeutic compounds are delivered to biofilms using conventional drug delivery methods (e.g., oral, inhalation, or injection) [23]. Thus, scientists seek to overcome these limitations using various approaches. Some examples include delivering drugs to biofilms using nanoparticle systems, interfering with bacterial communication and signaling pathways using small molecules, and inhibiting or degrading the EPS matrix [24]. Co-delivery of different antimicrobial agents and/or antibiotics and antimicrobial adjuvants within one nanocarrier system is another significant advantage, achieving a synergistic therapeutic effect and leading to potentially improved antimicrobial activity.

Various nanocarriers have been used to deliver a wide range of antimicrobial agents, i.e., polymeric nanoparticles, mesoporous silica nanoparticles, hydrogels, dendrimers, and lipid nanoparticles [24]. In this regard, lipid nanocarriers (LNCs) are considered attractive antimicrobial carriers owing to their biocompatibility and versatility, and they have also been demonstrated to improve the efficacy of existing antibiotics [25,26,27,28,29]. As LNCs are biomimetic and biocompatible drug delivery systems, they can be utilized to overcome the diverse physical, biological, and chemical barriers of bacteria. They can also improve absorption, permeability, bioavailability, and biofilm targeting to enhance the efficacy and decrease the toxicity of antimicrobial agents. Lipid nanocarriers can also improve antibiotic penetration across physical and chemical barriers, fuse with bacterial cell membranes, provide a stimuli-responsive release, and synergize the activity of loaded antibiotics and adjuvants to improve overall antimicrobial efficacy [30,31,32]. 

This review focuses on the treatment of biofilms with lipid nanocarrier-enabled delivery of antibiotics and antimicrobial adjuvants. This has not been reviewed before, highlighting the novelty of this review. The application of lipid nanocarriers (LNCs) as an anti-biofilm therapy is highlighted, with examples from recently published studies. The mechanisms of biofilm formation and challenges associated with biofilm eradication leading to suboptimal clinical treatment are discussed in detail. Insights into different types of LNCs for the delivery of antibiotics and antimicrobial adjuvants are also provided, together with an overview of recent advances in the application of potential anti-biofilm agents.

## 2. Biofilms—Definition, Composition, Life Cycle, and Therapeutic Challenges

Biofilms can be defined as a community of microorganisms that are embedded in an extracellular polymeric substance (EPS). The EPS primarily consists of polysaccharides, lipids, proteins, and nucleic acids (extracellular DNA (eDNA) and RNA) that constitute extremely hydrated polar mixtures, which adds to the overall scaffolding and three-dimensional structure of biofilms [7]. The composition of a mature biofilm is ~5–25% bacterial cells and 75–95% EPS [33]. The EPS usually has a thickness ranging from 0.2–1.0 μm and is present in both Gram-positive and Gram-negative bacteria. The EPS utilizes electrostatic forces, van der Waals forces, and hydrogen bonding to achieve the adhesion and cohesion of biofilms to solid surfaces, and it also contributes to biofilm maturation [10]. Biofilms are viscoelastic, and the EPS offers physical support against chemical and mechanical stresses [34]. The EPS also protects attached cells, thus decreasing the effects of antibiotics or antimicrobial agents [35]. The EPS layer aggregates antimicrobial molecules of up to 25% of its weight and limits the transport of biocides through its adsorption sites [36]. In addition to bacteria, biofilms comprise > 70 % water by weight, which facilitates storage and transport of auto-inducers (AI), nutrients, and waste products [37].

The formation of biofilms is a complex and endless cycle involving multiple parameters, but we can generally categorize it into five stages (shown in Figure 1) [38]. These include: (I) attachment, when microbes reversibly adsorb to surfaces through weak interaction (i.e., van der Waals forces) with abiotic or biotic surfaces; (II) colonization, when microbes irreversibly attach to the surfaces through stronger hydrophobic/hydrophilic interactions using pili, flagella, exopolysaccharides, lipopolysaccharides, and collagen-binding adhesion proteins; (III) development, when multiple layers of cells proliferate and accumulate, producing and secreting EPS; (IV) maturation, describing the formation of a stable three-dimensional community which comprises channels for the efficient distribution of nutrients and signaling molecules within the biofilm; (V) active dispersal, when due to the interactions between intrinsic or extrinsic factors, microbial cells clump or detach, and the dispersed cells subsequently colonize elsewhere [38].

Various conditions in biofilms promote the development of antibiotic resistance, through high mutation rates and the presence of persister cells, which survive antibiotic therapy owing to tolerance in a slowly growing population, and antimicrobial selective pressure. In non-growing, nutrient-deficient bacterial populations, activation of adaptive stress responses (oxidative stress, stringent response, SOS, or RpoS) causes adaptive mutagenesis, which is consistent with rapid growth of populations situated in the outermost layer of biofilm, promoting higher mutagenesis in biofilms [8,39].

Recently, different hypotheses have been suggested to describe the mechanisms of antimicrobial resistance in biofilms (Figure 2) [26,40]. The first hypothesis states that there is delayed penetration of antibiotics into biofilms. There is some evidence for this hypothesis, i.e., the absence of a generic barrier for the diffusion of antibiotic-sized solutes through the EPS [41]. It is noteworthy that the enzymatic inactivation of antibiotics in biofilms can lead to a delay in antibiotic penetration [42]. A low diffusion of antimicrobial agents within the EPS matrix also affects biofilm survival; e.g., β-lactam antibiotics at sub-MIC concentrations escalate alginate synthesis in *P. aeruginosa* biofilms and enhance the mucus-producing coagulase-negative staphylococci levels in biofilms [43]. Furthermore, limited penetration can be attributed to the adsorption of antibiotics on the EPS owing to the binding of positively charged antibiotics such as aminoglycosides to the negatively charged EPS [44,45].

The second hypothesis is based on changes in the chemical microenvironment inside the biofilm. Studies have shown that oxygen could be depleted across layers of biofilms, thus forming anaerobic conditions in deeper layers. Another significant difference among the microenvironments of planktonic bacteria and biofilms is associated with the pH difference in the bulk fluid and inside the biofilm, resulting in altered antibiotic activity [46]. In addition, changes in the osmotic environment within the biofilm may induce osmotic stress responses and lead to changes in antibiotic susceptibility, reducing the permeability of the cell envelope to antibiotics [24]. Studies have demonstrated that aminoglycoside and β-lactam antibiotics have a decreased efficacy against the same bacteria in anaerobic media compared to aerobic media [47]. Under moderate aeration conditions, bacterial cells showed higher resistance to both antibiotics. Free microorganisms were extremely resistant to antibiotics under anaerobic conditions. The researchers concluded that agar-embedded bacteria were significantly less sensitive than suspended cells under hypoxic conditions and suggested that this effect was related to the limited uptake of antibiotics by hypoxic cells, specifically due to the thickness of the biofilm [43]. At the same time, this layer exhausts substrates while accumulating inhibitory waste products, causing bacteria to enter a dormant state (persister cells). The reduced metabolic state protects the pathogens from antibiotic action, as most antibiotics rely on actively growing and rapidly dividing cells [48]. For example, in slow-growing *E. coli*, the expression of penicillin-binding protein (PBP) is negligible. Therefore, antibiotics like ceftriaxone and ceftazidime are less effective, irrespective of the presence or absence of growth-limiting nutrients [49]. Moreover, these persister cells are thought to be responsible for biofilm reseeding when antibiotic treatment is discontinued in clinical settings [50]. 

The third hypothesis of antibiotic resistance is associated with microbial subpopulations with distinct genotypic and phenotypic states within the biofilm matrix. Such microorganisms exhibit different antibiotic susceptibilities relative to persister cells and thus have unique properties and are greatly resistant to antibiotics [40]. The fourth hypothesis relates to the presence of a strong quorum-sensing system within biofilms. Bacteria communicate via quorum sensing (QS); through the secretion of signaling molecules (i.e., autoinducers (AI)) that are synthesized and function according to their density in a constrained environment. They respond through the activation of specific gene products, including enzymes, toxins, and virulence factors. QS can also affect biofilm development, the expression of virulence genes, biofilm resistance to antimicrobial treatment, and the induction of inflammatory responses [51]. This QS system is present in both Gram-positive and Gram-negative bacteria. Interestingly, it has been found that the diffusion of signaling molecules in the biofilm is not restricted. Within the biological layer, signaling molecules migrate shorter distances, facilitating cellular communication and reaction to these molecules [52]. 

Moreover, inactivation of the activity of antimicrobial agents by the immune system is another therapeutic challenge to address. Therefore, a strategy to mask antimicrobial agents from the immune system is required [53]. Yet, even if an antimicrobial agent successfully penetrates the biofilm, the acquired resistance of the bacterial cells to formerly consumed antibiotics can compromise the antimicrobial agent’s efficacy. This renders many currently utilized antimicrobial agents ineffective [53,54]. Therefore, promising antimicrobials against biofilms are characterized by biocompatibility, stability, non-immunogenicity, selective targeting, and biofilm penetration. Particularly, nano drug delivery systems are among the most desirable antimicrobial drug delivery systems because of their tunable size, stability, biocompatibility, and easy surface functionalization, making them promising candidates for antimicrobial delivery to biofilms [24].

## 3. Various Approaches to Combat Biofilms

Several approaches are being developed to treat biofilms. Biofilms can be treated by using either antibiofilm agents that target the different compounds that are responsible for biofilm formation (Figure 3A) or therapeutics that directly target the process of biofilm formation (Figure 3B). For a comprehensive review of these approaches to treating biofilms, readers are encouraged to read the review by Shrestha et al. [55]. 

However, it is imperative to recognize the challenges associated with the use of these anti-biofilm agents/techniques, i.e., their limited performance in vivo, their cytotoxicity to host cells, and their potential to induce resistance in biofilms. Promising next-generation anti-biofilm strategies depend on a multi-pronged approach, benefiting from recent advances in synthetic biology, nanotechnology, and antimicrobial drug discovery. Despite recent advances, the shortcomings of current and future anti-biofilm agents necessitate further research for their safe and effective clinical translation [56]. Translating complex anti-biofilm agents from controlled in vitro or in vivo analogous settings to real clinical settings requires a collaborative, multidisciplinary effort.

A combined approach using anti-biofilm drugs with different modes of action and biofilm-targeted immunotherapies may provide the following benefits: (i) simultaneously degrading the EPS, inducing biofilm dispersion, and eliminating persister cells, thus significantly improving the eradication of established biofilms, and (ii) overcoming antimicrobial resistance arising from the use of antibiotics alone. Additionally, the development of safe and on-demand antibiofilm drug delivery systems is crucial to avoid overdosing, which may increase the cytotoxicity to the host or development of AMR in biofilms. 

### 3.1. Antimicrobial Adjuvants to Improve Biofilm Killing

Multi-target action of antimicrobial agents can in principle be achieved through physical combinations of different compounds. Combination therapies were explored soon after the discovery of antibiotics, but the mechanisms of action were not well understood. By the mid-1950s, more than 60 combinations (two components or more) were identified. Early combined use of antibiotics improved the efficacy of sulfonamide, trimethoprim, penicillin, and streptomycin [57,58]. The significance of combination therapy in the treatment of tuberculosis and leprosy was recognized in the 1950s and 1960s, respectively [59]. Combination therapies are still used today, supported by systematic and clinical data, and are used based on clinical considerations [20]. 

In the context of bacterial biofilms, one of the most promising approaches is to combine antibiotics with adjuvants that do not interfere with the pathways necessary for bacterial growth and viability and are therefore less likely to induce resistance. Adjuvants are compounds that are co-administered with antibiotics to improve their antimicrobial activity [60]. These agents have multiple modes of action, such as: (i) improving bacterial membrane penetration, (ii) inhibiting biofilm formation and/or the formation of virulence factors and elements of antibiotic resistance, (iii) blocking antibiotics efflux pumps, and (iv) changing the phenotype through biofilm dispersion from the biofilm to the planktonic form [61]. Antimicrobial adjuvants have emerged as a promising approach to combat AMR and restore the efficacy of existing antibiotics but have currently only been narrowly explored. Currently, β-lactam/β-lactamase combinations are the only approved fixed-dose antibiotics and adjuvant combinations [20]. There is a need to re-evaluate antimicrobial monotherapy and investigate the use of antimicrobial adjuvants to reduce the development of AMR. 

Recently, IDR-1018 (the innate defense regulator peptide-1018, a 12-mer cationic peptide) has been introduced as a new class of antibiotic adjuvant [62]. IDR-1018 has not only demonstrated broad-spectrum activity but also exhibited synergistic effects with some commonly used antibiotics, i.e., ciprofloxacin, tobramycin, and ceftazidime, in 50% of assessments. It also reduced the antibiotics concentrations required for treatment against various bacterial biofilms by 2–64-fold [63]. A combination of peptide 1018 and antibiotics or each compound alone was tested. In all cases, conventional antibiotic treatment alone did not clear the preformed biofilm nor significantly induce biofilm cell death (Figure 4). Treatment with low concentrations of antibiofilm peptide 1018 alone resulted in reduced biofilm thickness, disruption of overall biofilm structure, and induced some cell death (Figure 4). These effects were significantly improved in the presence of low concentrations of antibiotics, while the antibiotics themselves did not affect preformed biofilms (Figure 4). For instance, combined treatment with peptide 1018 and ceftazidime completely eliminated mature biofilms formed by *A. baumannii* (Figure 4). The same antimicrobial combination at lower concentrations disrupted mature *S. aureus* MRSA biofilms and resulted in cell death (Figure 4). Peptide 1018 combined with tobramycin cleared biofilms of *K. pneumoniae* and killed biofilm cells of *E. coli* O157 (Figure 4). Likewise, treatment of *P. aeruginosa* PA14 mature biofilms with peptide 1018 and ciprofloxacin produced very small microcolonies composed of dead cells (Figure 4). On the contrary, treatment with ceftazidime and peptide did not clear mature biofilms formed by *Salmonella enterica*; but it did reduce biofilm thickness and resulted in greater cell death (Figure 4). Thus, we can conclude that peptide 1018 effectively potentiates antibiotic action when used in combination with conventional antibiotics, both to prevent biofilm formation and to treat mature biofilms formed by multidrug-resistant pathogens.

Moreover, in flow cell biofilm studies, the combination of low sub-inhibitory levels of ciprofloxacin (40 ng/mL) and peptide 1018 (0.8 μg/mL) reduced dispersal and caused cell death in mature *P. aeruginosa* biofilms. PCR studies showed that the peptide inhibited the expression of different antibiotic targets in biofilms. Therefore, treatment with this peptide characterizes a new strategy to enhance antibiotic activity against biofilms formed by multidrug-resistant pathogens. The IDR-1018 chemical structure has been illustrated in Figure 5.

The improved antimicrobial activity with adjuvants is achieved by directly inhibiting bacterial resistance mechanisms or decreasing the minimum inhibitory concentration of antibiotics needed to kill bacteria, thus enhancing the antibiotics’ effects and allowing currently available therapeutic options to be retained [64]. Repurposing drugs or antimicrobial compounds that have become obsolete opens the opportunity to develop multiple analogs as antibiotic adjuvants [65]. To date, no drugs have been repurposed as antibiotics. However, many existing drugs have shown activity against bacterial pathogens in vitro and are therefore termed “non-antibiotics”, or more precisely, “antimicrobial adjuvants”, as they potentiate the activity of antibiotics. Therefore, the drugs that are currently approved or in development for non-antibiotic indications may have antibiotic properties and thus may have the potential for repurposing, either alone or in combination with antibiotics [65].

Since different compounds simultaneously possess different mechanisms, the combined use of antimicrobials is a strategy that can improve the efficacy and spectrum of existing antimicrobials against pathogenic microorganisms. Furthermore, the combination of antibiotics and adjuvants has the potential to lower bacterial mutation rates and reduce resistance development due to putative bacterial target conservation. This strategy represents a synergistic approach in which the combined effects of adjuvants and antibiotics is greater than the sum of their individual effects, which can aid in reducing microbial resistance using lower doses of both agents [60]. 

#### 3.1.1. Quorum Sensing Inhibitors

Bacteria in biofilms communicate through quorum sensing (QS) via the secretion of chemical signaling molecules (i.e., autoinducers (AIs)) to instruct and accomplish colony behavior upon reaching a critical population density (i.e., swarm density) [66,67]. The concentration of AI increases with an increase in the bacterial population and results in altered gene expression when bacteria respond to these AIs. Quorum sensing comprises a two-factor signaling process that is distinct for Gram-positive and Gram-negative bacteria. In Gram-negative bacteria, the QS system comprises at least two regulatory proteins, namely LiR and LuxI. These proteins bind to protein receptors on the bacterial cell membrane. The signaling molecule binds to the receptor protein following internalization in the cell. The LuxI protein is involved in the synthesis of acyl-homoserine lactone (AHL) (Figure 5), which is used as a signaling molecule. AHL concentration rises with increasing cell population density. LuxR proteins are liable for binding to related AHL AI at threshold concentrations; these complexes also activate target gene transcription [68]. AHL production has not been observed in Gram-positive biofilms; however, the use of small peptide signaling molecules, i.e., autoinducing peptides (AIPs), that undergo post-translational processing, has been observed. These peptide signals act together with the sensor element of the histidine kinase two-component signal transduction system. The development of bacterial competency in *Streptococcus pneumoniae* and *Bacillus subtilis*, virulence in *Staphylococcus aureus*, and conjugation in *Enterococcus faecalis* are modulated by QS systems [34]. Non-species-specific autoinducer 2 (AI-2) (Figure 5) affects both Gram-positive and Gram-negative bacteria. Diffusible signaling molecules, e.g., cis-unsaturated fatty acids, also affect both Gram-positive and Gram-negative bacteria and are being explored for their potential therapeutic uses [69]. 

Quorum sensing inhibitors (QSIs) and quorum quenchers (QQ) have been recommended as anti-biofilm agents to impede the initial adhesion and formation of successive biofilm communities by specifically interfering with these processes [70]. Quorum quenching (QQ) refers to interference with QS through signal degradation such as the AHL-degrading enzymes acylases and lactonases [69]. Numerous QQ enzymes and compounds have been investigated. Most QQ molecules are of natural origin [71]. However, natural or synthetic QSIs have not shown clinical efficacy as monotherapies. This may be attributed to many factors, including the diversity of QS systems and the failure of QSI to penetrate the biofilm matrix effectively. Recently, the QS inhibitory potential of *Natrinemaversi forme* and ethyl acetate from cell-free supernatants against *P. aeruginosa* biofilms has been reported [72]. Many plant-based natural QS inhibitors have also been discovered [73] and are suggested to be promising anti-biofilm agents in the future [74]. These antibiofilm agents disrupt the QS system primarily in two ways: (i) inhibition and degradation of signaling molecules, and (ii) mimicking signaling molecules to inhibit their binding to the corresponding receptors [75]. Ajoene, a sulfur-rich molecule from garlic, reduces the expression of small regulatory RNA (sRNA) in *S. aureus* and *P. aeruginosa* and was the first compound identified as a broad-spectrum QSI, i.e., reducing RNAIII expression in *S. aureus* [76] and RsmY and RsmZ expression in *P. aeruginosa,* respectively [77], thereby inhibiting the translation of the EPS polysaccharides Pel and Psl and the type VI secretion system T6SS. In *P. aeruginosa*, T6SS causes the expression of different virulence factors and is closely related to the pyocyanin production, biofilm formation, and the pathogenicity of the bacteria [78]. These findings imply that the T6SS can be a potential therapeutic target to combat *P. aeruginosa* infection. In another study, it was found that ajoene reduced regulatory RNA and RNAIII expression in *S. aureus* and inhibited the expression of RNAIII-dependent virulence factors, e.g., protease, lipase, and α-hemolysin [77]. 

Human Cathelicidin LL-37, an anti-biofilm peptide, affects bacterial cell signaling systems and inhibits the biofilm formation of *P. aeruginosa* by downregulating QS system genes [79]. AMP interacts with bacterial membranes and in turn activates genes regulated by the QS system. These QS autoinducers cross the plasma membrane through membrane vesicles. As a result, this process activates the QS-related virulence gene expression. Emodin, an anthraquinone derivative isolated from *Polygonum cuspidatum* and rhubarb, efficiently downregulates the luxS gene in *Streptococcus suis* [80] and agrA, sarA, and icaA genes in *S. aureus* [81]. Autoinducers contribute to interspecies signaling. One remarkable autoinducer is the small autoinducer peptide molecule (AIP) from Lactobacillus species, which inhibits microbial growth and the production of bacterial toxins. Through the inhibition of exotoxin production, they intervene in the agr QS system [82]. However, QQs could be washed away during biofilm formation, which limits the use of these QS inhibitors in biofilms [83]. Therefore, a combination approach employing these QS inhibitors combined with antibiotics yields a new treatment strategy. For instance, a QSI, 3-amino-7-chloro-2-nonylquinazolin-4(3H)-one (ACNQ) combined with ciprofloxacin was co-encapsulated in alginate nanoparticles, exhibiting complete eradication of 24-h *P. aeruginosa* biofilm infections in an ex vivo 3D skin infection model [61].

Quorum quenchers (QQs), on the contrary, are often species-specific. Hence, a combination of quenchers is needed to eradicate mixed-species biofilms. Numerous QQ enzymes and compounds have been investigated. The metalloprotein AHL-lactonase in endophytic Enterobacter cell-free extracts was shown to degrade N-AHL, thereby considerably inhibiting *Aeromonas hydrophila* biofilm formation [84] Similarly, *Lactobacillus scleroderma* ZHG 2-1 demonstrated degradation of N-butyryl-dl-homoserine lactone (C4-HSL) and N-3-oxododecanoyl-dl-homoserine lactone (3-oxo-C12-HSL) and acted as anti-biofilm agent to combat *P. aeruginosa* biofilm [84]. A recent study exhibited demonstrated that catheters coated with amylases and acylases inhibited biofilm formation [85]. This coating demonstrated greater efficacy against Gram-negative bacteria, while a similar approach has been presented to combat *S. aureus* biofilm via QS targeting [86]. Further studies on AI2-based quorum quenching with brominated furanone have been proposed [69]. 

Unfortunately, due to the diversity of QS systems that regulate biofilm growth and dispersion, it is unlikely that molecules modulating specific signaling pathways can be employed as broad-spectrum biofilm dispersants. Additionally, initial evidence implies that therapies targeting known signaling factors across multiple species may disrupt human microbiota homeostasis [87]. The effects of QSIs on signaling factors in eukaryotic mammalian cells should also be cautiously monitored. Several in vitro and in vivo studies have established a link between N-acyl-homoserine lactones (AHLs) and the initiation of pro-apoptotic and pro-inflammatory responses, comprising direct disruption of regenerative processes [88,89]. However, even though these issues cause significant impediments to the further advancement of QSIs, they do not lessen the significance and potential of this new approach in combating a narrow spectrum of clinical biofilm infections, especially when combined with the continued understanding of bacterial cell-to-cell signaling networking and innovative drug delivery strategies. Some potential QSIs and QQs and their respective applications are summarized in Table 1.

#### 3.1.2. EPS-Degrading Enzymes

EPS degradation leads to biofilm dispersion with reduced bacterial protection, resulting in improved antibiotic activity. Enzymes play a key role in dispersing EPS matrix/biofilms, through which microorganisms become planktonic leading to improved killing at low doses of antibiotics. These enzymes can be produced naturally, synthetically, or recombinantly. These EPS-degrading enzymes are degrading proteins targeting substrates present in the biofilm matrix [110]. These enzymes can be employed both as a preventative therapy through inhibition of the production of biofilm matrix and as a therapeutic to disperse established biofilms. These enzymes do not possess intrinsic antimicrobial properties but when co-administered with antibiotics, they can target and eliminate biofilm-associated infections. Recent studies have demonstrated that various enzymes can effectively disrupt biofilm structure, including glycosidases (e.g., DspB, PgaB, Ps1G, Pe1A), proteases, lactonase, α-amylase lyase, and deoxyribonucleases (DNase I, rhDNase) (summarized in Table 2 with their potential applications). The most common mechanism of these enzymes is the inhibition of intercellular adhesion and dispersal of cell aggregates [111]. 

##### Glycosidases

Dispersion B (DspB)

Dispersion B (DspB), a glycoside hydrolase enzyme, is produced by *Actinobacillus actinomycetemcomitans* and hydrolyzes poly-β-1,6-N-acetyl-d-glucosamine (PNAG), an essential adhesion molecule required for the formation of biofilm and integrity in *Staphylococcus* and *E. coli* (including *E. coli* K-12), and clinical isolates [141,142]. In the EPS matrix, bacteria produce polysaccharides that promote their virulence, colonization, and survival [143]. Degradation of polysaccharides significantly weakens biofilms and makes the sessile microbial population more accessible to antibiotics leading to enhanced clearance of microorganisms from an infection. DspB has been shown to inhibit the biofilm formation and disperse biofilms of different bacteria like *E. coli*, *A. baumannii*, *A. actinomycetemcomitans*, *S. aureus*, *S. epidermidis*, *K. pneumoniae*, *A. pleuropneumoniae*, *Burkholderia spp*., *P. fluorescens*, and *Y. pestis* without presenting any antibacterial activity [143]. However, the combined use of DspB with antibiotics rendered bacteria more susceptible to antibiotic killing by inhibiting biofilm formation or by disrupting the pre-formed biofilms [144]. 

Ghalsasi and Sourjik engineered an *E. coli* strain to synthesize and secrete DspB into the microbe that successfully disrupted the preformed *E. coli* biofilms [144]. Combined treatment with tobramycin and DspB decreased bacterial number in *S. aureus* biofilms by 7500-fold, while tobramycin alone reduced cell numbers by only 40-fold [117]. DspB can be formulated into gel, and topical sprays, and permeated into medical plastics such as catheters to reduce hospital-acquired infections [143]. An innovative antibiofilm nanovector drug delivery system, consisting of DspB-permethylated-β-cyclodextrin/ciprofloxacin adamantyl was designed, which showed excellent antibiofilm activity against *S. epidermidis* biofilms [145]. When DspB was co-administered with teicoplanin in a catheter lock solution, it disrupted *S. aureus* biofilms, thereby enhancing the bloodstream infection’s elimination rate in intubated sheep [146]. Furthermore, the triclosan and DspB combination enhanced the eradication of *S. aureus*, *E. coli*, and *S. epidermidis* biofilms as compared to control, DspB alone, or triclosan alone, thus demonstrating synergistic anti-biofilm activity of the combination treatment [115]. A wound gel consisting of DspB, the antibacterial peptide KSL-W, and Pluronic^®^ F-127 was developed by Kane Biotech Inc. that significantly enhanced wound healing compared to controls in both infected and non-infected wounds [118]. However, all these studies conducted only efficacy studies, and no further stability or toxicity data for the enzyme were reported.

PgaB

PgaB is a novel recombinantly produced glycoside hydrolase, which is in the initial stages of preclinical development for the treatment of PNAG-dependent biofilms. PgaB is a two-domain periplasmic protein comprising a C-terminal PNAG-binding domain and an N-terminal deacetylase domain critical for export [147]. Yet, the precise function of the C-terminal domain of PgaB is not clear. Endogenously, PNAG is deacetylated by PgaB, causing a cationic charge to the exopolysaccharide, which is essential for biofilm adhesion and formation [111]. PgaB is catalytically different from DspB as DspB is both an exo and endo enzyme and cleaves the final polysaccharide unit and those within the polymer, whereas PgaB is specifically an endo enzyme i.e., cleaves within the polysaccharide unit [143]. Dustin et al. demonstrated that the PgaB C-terminal domain produced by *E. coli* and *B. bronchiseptica* can cleave dPNAG and is accountable for the glycoside hydrolysis of PNAG-dependent biofilms produced by *E. coli*, *S. carnosus*, and *S. epidermidis* and enhanced gentamicin mediated bacterial killing. The EC50 of the glycoside hydrolase domain for biofilm dispersion was 6 nM for clinical isolates of *S. epidermidis* biofilms [120].

Alginate lyase (AL)

Alginate lyase (AL), in addition to dPNAG hydrolases, can effectively disperse mature biofilms [148]. AL catalyzes the degradation of alginate and has been purified from a variety of organisms with diverse substrate specificities i.e., algae, terrestrial and marine bacteria, marine molluscs, and certain fungi and viruses [149]. For optimum biofilm dispersion, the substrate should match the specific enzyme’s activity. In well-matched enzyme and substrate experiments, AL reduced the viscosity of alginate-containing sputum in vitro [150,151]. *P. aeruginosa* can produce alginate polysaccharides, a significant component in the biofilm matrix structure of this bacteria protecting it from dehydration, and antimicrobial activity [152]. In mucoid strains, alginate is secreted in the surrounding medium and is not covalently bound to the cell surface. Hence, the presence of alginate-degrading enzymes is expected to enhance the susceptibility of *P. aeruginosa* biofilms. This objective can be accomplished by adding enzymes to the environment surrounding the bacteria and enhancing the expression of genes encoding alginate-degrading enzymes [153]. A recent study demonstrated that purified marine alginate lyase (AlyP1400) dispersed the *P. aeruginosa* biofilm and improved the tobramycin’s bactericidal activity [154]. 

Ps1G and Pe1A 

PslG is a periplasmic glycoside hydrolase encoded by the Psl exopolysaccharide biosynthetic operon [155]. PslG having a soluble, catalytically active glycoside hydrolase domain, can hydrolyze Psl in *P. aeruginosa* biofilms after removing the N-terminal transmembrane domain. PslG inhibited biofilm formation of environmental and clinical P. aeruginosa isolates within 24 h and was also able to disrupt freshly formed biofilms but had lesser activity in dispersing mature biofilms. Furthermore, PslG enhances the antimicrobial efficacy of colistin [123]. In full-thickness *P. aeruginosa*-infected wounds, PslG and tobramycin significantly decreased bacterial load by 1-log as compared to antibiotic alone at 72 h post-infection [156]. PslG is non-cytotoxic and supports immune defense; the enzyme does not modify host cell morphology and improves neutrophil-killing activity [123].

PelA is also a periplasmic glycoside hydrolase encoded in the Pel exopolysaccharide biosynthetic operon and comprised of a minimum of two catalytic domains—a CE4 deacetylase domain and a putative glycoside hydrolase domain [111]. In a study to examine the glycoside hydrolase activity of Pe1G, the N-terminal domain of PelA was eliminated to generate the PelA47-303 construct (termed PelAh), expressed, and purified [155]. Prophylactic therapy with PelA demonstrated a 2.5-log decline in *P. aeruginosa* CFUs, while the treatment of pre-formed biofilms with Pe1A exhibited significant biofilm dispersion within 24 h. Moreover, biofilm disruption by PelA is insensitive to the mature biofilms [123]. PelAh also enhanced colistin’s efficacy and neutrophil killing by approximately 50% [123].

##### Deoxyribonucleases

eDNA is an essential structural component of the EPS matrix of the bacterial biofilm, forming a lattice-like structure like Holliday junctions. Regardless of the significance of eDNA in bacterial biofilms, it did not receive widespread consideration until 2002 when Whitchurch et al. demonstrated increased bactericidal efficiency by exogenous addition of DNase I, which in combination with antibiotics disperses biofilms [157]. Since then, considerable work has been done to eradicate biofilm infections using various DNases targeting eDNA.

RhDNase has been recognized as a strong biofilm-dispersing enzyme in which eDNA degradation leads to the disruption of biofilm structure in highly tolerant *E. coli* and *P. aeruginosa* infections. Furthermore, it also decreases the antibiotic resistance of S. epidermidis and S. aureus biofilms [158]. An rhDNase-based drug Pulmozyme^®^ is marketed in France by Roche Laboratories which is a solution for nebulization used for treating *P. aeruginosa* infections in cystic fibrosis patients [159]. The drug demonstrated biofilm detachment activity against *S. pneumoniae* and *P. aeruginosa* biofilms in vitro. Recent studies have revealed that DNase I has broad compatibility with several antimicrobial drugs like silver sulfadiazine, ceftazidime, and proteinase K [133,160,161]. Treatment of antibiotic-resistant biofilms with DNase I enhanced matrix permeability, leads to a subsequent enhancement in antibiotic susceptibility [162]. Baelo et al., synthesized DNase I-coated ciprofloxacin-loaded PLGA nanoparticles for biofilm treatment. This combination was effective against *S. aureus* and *P. aeruginosa* biofilm inhibition and removed >99.8% of the established biofilms [134]. However, these nanoparticles pose some disadvantages i.e., low loading capacity. Moreover, DNase I-coated nanoparticles do not possess an antibacterial effect; therefore, they still need to be used in combination with other antimicrobials. Therefore, the use of carrier formulations for biofilm-dispersing enzyme and antibiotic combinations that enable protection from protease degradation, limited interference with enzyme activity, and controlled release warrants exploration. 

##### Proteases

Extracellular proteins are the main components of EPS, accounting for a large portion of biofilm biomass, and are critical for microbial maintenance (surface adhesion, cell aggregation, structural integrity), and modification of EPS [163,164]. Enzymatic degradation of EPS extracellular proteins is an efficient way for biofilm eradication. To date, many proteases capable of dispersing biofilms have been discovered and studied. Several proteases that facilitate biofilm dispersion have been recognized e.g., the serine protease Esp secreted by a subpopulation of *S. epidermidis*. Purified Esp inhibited biofilm formation and disrupted pre-formed S. aureus biofilms, enhancing the sensitivity of biofilm-embedded S. aureus to human beta-defensin 2 (hBD2), an antimicrobial peptide component of the innate immune system of human [165]. Another study showed that proteinase K (2 μg/mL) efficiently inhibited the biofilm formation of bap-positive *S. aureus* V329 and other *S. aureus* isolates (SA7, SA10, SA33, SA352), and significantly improved the efficacy of gentamicin against all *S. aureus* biofilm [166]. Cysteine proteases secreted by equine mesenchymal stromal cells (MSCs) can disrupt *methicillin-resistant Staphylococcus aureus* (MRSA) biofilm, thereby enhancing the efficacy of the antibiotics formerly tolerated by the biofilms [167]. 

## 4. Lipid Nanocarriers Mediated Delivery of Antibiotics and Antimicrobial Adjuvants

The efficacy of anti-biofilm agents can be improved by various approaches including encapsulation into nanoparticles for optimum delivery or combination of various drugs to enhance antibacterial activity. Yet, the cytotoxicity and in vivo therapeutic efficiency of anti-biofilm drugs remain critical issues. Furthermore, these issues are exacerbated when antibiotics or antimicrobials are delivered without a carrier. Nanostructures serve as a multifunctional platform that can be specifically designed to co-deliver different antimicrobial agents that specifically target biofilm cells without affecting host cells [56].

Recalcitrant infections caused by biofilms cannot be effectively treated with antibiotic therapy based on conventional pharmaceutical formulations. Over the past decade, lipid nanocarriers (LNCs) have drawn increasing attention as alternative methods to deliver a variety of compounds like proteins and peptides, lipophilic and sensitive compounds associated with stability issues [24]. Encapsulation of antimicrobial agents into lipid nanocarriers offers several advantages such as prevention of inactivation, fusogenicity to enhance antimicrobial activity, targeted delivery due to tailorable surface to apply any type of targeting strategy, high drug loading, enhanced biodistribution, and pharmacokinetic profiles, as well as reduced adverse side effects and systemic toxicity [26]. Furthermore, the controlled and sustained release of antimicrobials from these nanocarrier formulations is advantageous to combat biofilms and for better antimicrobial activity [168]. Biofilm-nanocarrier interaction principally involves three steps that are responsible for better antibiofilm characteristics: (i) nanocarriers transport around the biofilm, (ii) nanocarriers attach to the biofilm EPS, (iii) nanocarriers penetrate the EPS and migrate in the biofilm through diffusion that may depend on the biofilm charge, pore size, hydrophobicity, and EPS chemical gradient [169]. 

LNCs are usually non-spherical in shape, either due to electrostatic interactions between the the polar/ionic phospholipid head and the solvent or owing to the presence of nonpolar aliphatic hydrocarbon moieties in the solvent [170]. These LNCs made from uniform lipid bilayers or solid cores, can encapsulate a variety of antimicrobials. Hydrophilic drugs can be entrapped in the aqueous regions, whereas lipophilic drugs can be entrapped in the lipid channels [171]. Antimicrobials-loaded LNCs have made great progress in the past decade, providing antimicrobial protection against chemical or enzymatic degradation, thereby maximizing drug interaction with target bacteria [25,172]. Therefore, these properties can improve the antimicrobial activity of antibiotics. Additionally, antimicrobial-loaded LNCs can inhibit biofilm formation, increase biofilm penetration, and enhance intracellular bacterial killing [173].

In recent years, research on lipid nanocarriers has flourished for the co-delivery of antibiotics and antimicrobial adjuvants, and the main categories include LCNPs, liposomes, solid lipid nanoparticles (SLN), and nanostructured lipid carriers (NLC), which are highly regarded in current research and clinical trials. Among them, LCNPs and liposomes, have superior advantages i.e., biocompatibility, controlled release, decreased immunogenicity, increased stability, and the ability to encapsulate hydrophilic and lipophilic agents simultaneously. Thus, co-encapsulation of antimicrobial agents in LNCs is an advanced approach that can retain the functional activity of two or more antimicrobial drugs having different natures, solubility, and properties, while improving their anti-biofilm activity. To facilitate clinical translation, more research on targeted co-delivery by lipid nanocarriers is required to enhance permeability, minimize toxicity, and retention effects, and reduce the shielding effect of the protein corona as recently limited studies are available for the combined delivery strategy. The advantages and detailed comparison of LNCs have been extensively reviewed elsewhere [38] and are out of the scope of this review. We have precisely discussed specific LNCs for their application in the co-delivery of various antibiotics and antimicrobial adjuvants (summarized in Table 3 and illustrated in (Figure 6).

LNCs first interact with the bacterial surface, driving nanoparticle antibacterial activity [25]. The mechanisms of LNCs against bacteria include destruction of cell walls and cell membranes, membrane fusion, and destruction of bacterial intracellular components. These mechanisms are influenced by LNC composition, particle size, zeta potential (ZP), and large surface area to volume ratio as well as antibiotic drug loading [25,173]. In addition, van der Waals forces, electrostatic attractions, and hydrophobic interactions promote the interaction of multiple LNCs with bacterial cells [174]. LNCs can be formulated for targeting specific bacteria, using lipids and surfactants that favor increasing membrane permeability to maintain sufficient antibiotic concentrations, leading to membrane rupture [25]. For instance, cationic LNCs can promote the aggregation of nanoparticles around bacteria through electrostatic interactions, thus increasing the antibiotic concentration at the site of action [28]. 

Particle size is a key property of LNCs that determines their journey behavior in biological systems. Particles smaller than 100 nm mainly cross biological membranes through the endocytic pathway [175], and particles smaller than 10 nm are easily cleared by glomerular filtration [176]. On the contrary, drugs above 100 nm have a shortened half-life in the bloodstream due to the active involvement of the mononuclear phagocytic system (macrophages and dendritic cells) when administered systemically [177].

Furthermore, zeta potential is another significant property that directly affects the stability of LNCs. Therefore, zeta potential can be used to optimize formulations, address particle surface modifications, predict interactions, and assess long-term stability. Generally, particles with a ZP higher than 30 mV in the module are deemed stable and do not aggregate due to the electrostatic repulsion between charged nanoparticles. In contrast, steric stabilization includes stabilization by nonionic surfactants and polymers [177]. These macromolecules adsorb on the surface of the nanoparticles, hindering aggregation and counteracting the attractive van der Waals forces [178]. 

The activity of LNCs against biofilms is affected by a diffusion coefficient, which is directly related to nanoparticle composition, particle size, and biofilm composition [179]. LNCs targeting biofilms displayed particle size ranges from 119–406 nm, PDI values < 0.3, and ZP ranges from −43 to +18 mV. The appropriate particle size for treating biofilms can be < 500 nm, with a preferred range between 5 and 200 nm [24,180]. Cationic LNCs exhibited better penetration, distribution over the entire negative biofilm surface, and reduction in biofilm integrity [181]. However, strong electrostatic interactions may retain nanoparticles on the biofilm surface, limiting their penetration [182]. In contrast, negative and neutral LNCs revealed potent anti-biofilm activity against *S. aureus*, *P. aeruginosa*, and *Klebsiella oxytoca* [182,183]. Therefore, the zeta potential of LNCs against biofilms remains poorly understood.

### 4.1. Liposomes

Liposomes were the first phospholipid vesicle systems established in the 1960s and consist of a phospholipid bilayer mimicking the plasma membrane of human cells which can be readily fused with microbes. Hence, liposomes demonstrate good biocompatibility and could improve drug distribution through plasma membrane [184]. The main structural component of liposomes is phospholipids, which form a spherical structure when mixed with an aqueous solution [185] (Figure 6). Another important component of liposomes, cholesterol, helps provide stability to the liposome structure and enhances drug solubility in the blood circulation system [186]. Liposome particle sizes range from 20 nm to >1 μm (cholesterol can produce large vesicles in the range of 0.025–2.5 μm), with a typical hydrophobic bilayer and hydrophilic core encapsulation structure. Therefore, liposomes can maintain and stabilize hydrophilic drugs in the aqueous core and trap lipophilic drugs in the lipid bilayer, thus contributing to their multifunctionality [187].

Liposomes have been investigated in various studies as carriers of numerous anti-biofilm agents [24]. To allow improved permeation and targeting of antimicrobial agents to biofilms that are not normally reached, the diameter of liposomes should favorably be between 100–200 nm [178]. The internal hydrophilic core of the liposomes offers an appropriate environment for several conventional antibiotic encapsulations, whereas fusion of the liposomes with phospholipid membrane causes enhanced cellular uptake of drugs releasing drugs inside the cell’s cytoplasm and hence improved killing of biofilms. This type of directed and targeted delivery is a well-known advantage of liposomes. Owing to the similar chemical composition of the liposomes to bacterial membranes, this fusogenicity of the liposomes only generates transport channels instead of membrane rupture. Penetration of the bacterial membrane is supposed to take place through these pores.

The structure of liposomes is suitable for encapsulating a variety of antimicrobial agents. Yet, these structures are thermodynamically unstable systems that are prone to aggregation or degradation, resulting in limited feasibility of antimicrobial encapsulation [188]. To overcome the limitations associated with liposome stability, different studies have suggested the use of biopolymers as coating materials. Generally, biopolymers can help stabilize particles by modifying the liposome’s surface through covalent or non-covalent interactions. Therefore, the incorporation of biopolymers, including proteins, polysaccharides, and their derivatives, signifies a promising strategy to improve the performance of liposomes, making them more stable, protected, and therefore more suitable [189]. Among the biopolymers that can be used to coat liposomes, besides starch, alginate, and pectin, chitosan is one of the most used biopolymers [190]. 

Cationic liposomes have greater potential to attach to the negatively charged EPS and penetrate better into biofilms. Therefore, improved bacterial killing is observed than unformulated antimicrobial solutions or their uncharged liposomal counterparts, signifying their stability in the liposomal formulations [31]. However, negative charges of the EPS matrix hamper the action of antibiotics and various drug delivery systems, adding to the complexity of targeting bacteria with simple electrostatic interactions. For instance, cationic antibiotics such as colistin sulfate and aminoglycosides bind to the negatively charged EPS matrix, immobilizing the antibiotics, and restricting their access to the encapsulated bacteria [191]. Thus, apart from fusion with bacterial membranes, the size and surface charge of liposomes could be tailored using different lipid combinations, several of which have been studied and extensively reviewed by Forier et al. from different aspects including stability [31]. Captivatingly, both negatively and positively charged liposomes have been tested, each with distinctive targeting arguments.

Moreover, liposome surface can be easily altered with specific ligands or molecules to actively bind the specific target sites hence leading to target specificity. Other potential mechanisms for improving anti-biofilm activity are enhancing drug interaction with bacterial biofilms, inhibiting bacterial growth, reducing the synthesis of virulence factors, and impeding the motility of drug-resistant strains [24]. However, due to the limited space in the liposome bilayer, it is hard to attain higher drug loading for hydrophobic drugs. It is crucial to attain a precise balance between high drug loading capacity and stability and particle size distribution of liposomes [170]. Hence, it is crucial to optimize the composition and characteristics of lipid bilayers and preparation methods.

All the above-mentioned characteristics allow liposomes to be used as effective nanocarriers for the encapsulation of antimicrobial compounds including antimicrobial adjuvants. Biofilm-dispersing enzymes can be entrapped either in the lipid bilayer and internal aqueous cavity or in the external environment. The catalytic activity and the substrate specificity of the enzyme are enhanced as the interaction of the enzyme with the lipid bilayer stabilizes the protein’s quaternary structure [192]. Enzyme stability offers a backbone strategy to exploit catalytic activity and reduce susceptibility to chemical and biological degradation such as from the proteases and environmental stimuli. Biofilm-dispersing enzymes when anchored in the outer shell of liposomes, can serve as a stabilizing unit or targeting ligand to guide the carrier through electrostatic interactions i.e., positive charge towards the negatively charged EPS matrix. Liposomes comprised of dipalmitoylphosphatidylglycerol (DPPG) and dipalmitoylphosphatidylcholine (DPPC) were stabilized by electrostatic binding of the lysozyme and loaded gentamicin [190]. The lysozyme-associated liposome potentiates the gentamicin efficacy against preformed *S. aureus* and *P. aeruginosa* biofilms by 4-fold in comparison to the gentamicin alone. Although the enzyme was located on the liposome’s surface, it maintained its stability in vitro, however, no in-vivo studies were performed to investigate the preclinical activity [190].

Thorn et al. have used liposomes and liquid crystal nanoparticles (LCNPs)-an emerging lipid-based drug delivery system (for a detailed discussion on LCNPs, please refer to Section 4.2), as comparative drug delivery systems for the co-delivery of glycoside hydrolase-Ps1G and tobramycin. The results demonstrated that liposomes significantly enhanced the efficacy of tobramycin compared to tobramycin alone or Ps1G and tobramycin as unformulated solutions but less compared to LCNP formulations. Thus, the therapeutic efficacy of the antibiotics against biofilms can be enhanced by delivering with liposomes but is inferior to LCNPs [13]. In another study, the potential of cationic liposomes encapsulated with DNase I and proteinase K (EE; 67–83%) has been investigated for the treatment of cutaneous and catheter infection through the eradication of preformed *Cutibacterium acnes* biofilm [193]. In vitro, porcine skin penetration demonstrated the facile delivery of cationic liposomes to the epidermis, deeper skin layers, and hair follicles. These liposomes further demonstrated promising in vivo activity in eliminating colonization of *C. acnes* in murine skin and catheters. There was a 2-log decline in colony-forming units (CFU) in catheters treated with liposomes compared to untreated controls.

A naturally occurring QSI, farnesol, was co-encapsulated with ciprofloxacin in a liposomal system to treat *P. aeruginosa* biofilm [194]. Four different liposome formulations i.e., Lcip + far (ciprofloxacin and farnesol); Lcip (Ciprofloxacin); Lfar (farnesol); Lcon (control) were developed by dehydration–rehydration method. The efficacy of developed liposomes was evaluated against 24 h *P. aeruginosa* biofilm qualitatively (XTT reduction assay and crystal violet assay) as well as qualitatively (transmission electron microscopy (TEM) and confocal laser scanning microscopy (CLSM)). Metabolism of biofilms was significantly reduced upon treatment with Lcip or Lcip + far compared with free ciprofloxacin (XTT, *p* < 0.05). Upon Lcip + far administration, the concentration of ciprofloxacin needed to attain similar inhibition of biofilm was 10-fold or 125-fold lower as compared to Lcip or free ciprofloxacin, respectively (*p* < 0.05). TEM and CLSM established that biofilms were mainly destroyed, with a higher proportion of dead cells and an enlarged depth of biofilm killing upon treatment with Lcip + far as compared to other liposome formulations. Therefore, co-delivery of ciprofloxacin and farnesol may be a promising strategy to combat resistant *P. aeruginosa* biofilms by improving biofilm killing at significantly lower ciprofloxacin doses [194].

In a different study, a liposomal system was successfully prepared with DMPC and cholesterol (2:1 molar ratio) to encapsulate different aminoglycosides e.g., tobramycin, gentamicin, and amikacin, and their antimicrobial activity was evaluated in combination with DNase, alginate lyase and N-acetylcysteine against two clinical isolated of *P. aeruginosa* (one mucoid and other non-mucoid) [150].

**Table 3 pharmaceutics-16-00396-t003:** LNCs investigated by various researchers for the antibiotics and antimicrobial adjuvants delivery to bacterial biofilms.

Nanocarrier	Encapsulated Agent	Class	Biofilm	Testing Method	Outcome	Ref.
Liposomes	Ps1G + Tobramycin	Biofilm-dispersing enzyme + Aminoglycoside antibiotic	*P. aeruginosa*	CV assay, MBEC assay	Improved the activity of tobramycin with a 20% reduction in the biofilm biomass.	[13]
Liposomes	Lysozyme + Gentamicin	Biofilm-dispersing enzyme + Aminoglycoside antibiotic	*P. aeruginosa* and *S. aureus*	CV assay, MTT assay	Liposomal formulation significantly reduced the biofilm biomass and live bacterial count compared to free drugs and enzymes.	[190]
Liposome	N-Acetylcysteine + Tobramycin	Antibiotic adjuvant + Aminoglycoside antibiotic	*Escherichia coli*, *Acinetobacter baumannii*, and *Klebsiella pneumoniae*	CV assay	Tobramycin and encapsulated liposomes significantly reduced the biofilm biomass and showed enhanced efficacy in inhibiting biofilm formation compared to unformulated drugs.	[195]
Liposome	N-Acetylcysteine + Azithromycin	Antibiotic adjuvant + antibiotic	*E. coli* (Clinical isolate)	CV assay	Azithromycin encapsulating in liposomes demonstrated higher biofilm reduction i.e., 93.22%) at 1X MIC.	[196]
liposome	Lysozyme + Chlorhexidine + Lactoferrin	EPS-degrading enzyme + antibiotic + glycoprotein	*Streptococcus mutans*, *Streptococcus sobrinus*	CFU enumeration	Encapsulated chlorhexidine completely inhibited the biofilm formation.	[197]
Liposome	Serratiopeptidase (SRP) + Levofloxacin	EPS-degrading enzyme + antibiotic	*S. aureus* infected rats	CV assay	Levofloxacin (sub-MIC concentration) co-encapsulated with SPR significantly eradicated the preformed biofilm i.e., >90%.	[198]
Liposome	DNase I and proteinase K	EPS-degrading enzymes	*Cutibacterium acnes*	CV assay, Porcine skin model (in-vitro), Murine skin and catheters (In vivo)	Dual enzyme-loaded liposomes exhibited greater biofilm-formation inhibition and deeper penetration (85%) into biofilm thickness. Enhanced penetration and facile delivery into porcine skin in-vitro. Potent in-vivo activity in eliminating colonization of *C. acnes* in murine skin and catheters with a 2-log reduction in CFU in catheters treated with liposomes compared to untreated controls.	[193]
Liposome	CDC and PF	Quorum sensing inhibitors	*P. aeruginosa*	CV assay	Liposomal formulations demonstrated dose-dependent anti-biofilm activity compared to fee QS inhibitors.	[108]
Liposome	2-nitroimidazole derivative, 6-NIH + DETA NONOate + Azithromycin	Antibiotic adjuvant + biofilm dispersant + antibiotic	*P. aeruginosa*	CV assay, CLSM	Liposomal formulation significantly eradicated mature biofilm, efficiently killed dispersed bacteria, inhibited the metabolism of survivors, and also inhibited recurrent infection by preventing bacteria from adhering to airway epithelial cells.	[199]
Liposome	Farnesol + Ciprofloxacin	QSI + Quinolone antibiotic	*P. aeruginosa*	XTT reduction assay, Confocal laser scanning microscopy (CLSM)	80% reduction of *P. aeruginosa* biofilm biomass at 0.128 µg/mL concentration of ciprofloxacin. Greater cell death was observed via CLSM imaging after the biofilm treatment with the formulation compared to the liposomal ciprofloxacin alone.	[194]
Liposome	Bismuth ethanedithiol (BiEDT) + Tobramycin	Antimicrobial adjuvant + antibiotic	*P. aeruginosa*	QS and Virulence factor assay (In vitro), Sprague Dawley rats (In vivo)	Encapsulated liposomes effectively disrupt the quorum sensing and significantly reduce the chitinase, lipase, and protease production (In-vitro) with a 3-log CFU reduction (In-vivo) compared to unformulated drugs.	[200]
Liposome	Bismuth-ethanedithiol + Alginate lyase + Tobramycin	Antimicrobial adjuvant + EPS-degrading enzyme + antibiotic	Mucoid *P. aeruginosa* (Clinical isolate)	MBEC assay, Carbazole assay	The anti-biofilm activity of bismuth-ethanedithiol + TOB compared with unformulated TOB was decreased by 4–32-fold. While addition of enzymes markedly increased the biofilm eradication.	[201]
LCNPs	Alginate lyase + Gentamicin	Biofilm-dispersing enzyme + Aminoglycoside antibiotic	Mucoid *P. aeruginosa* (Clinical isolate)	CV assay, MBEC assay	Infection-responsive antibiotic release, >2-log decline in *P. aeruginosa* biofilm compared to unformulated solutions.	[202]
LCNPs	Ps1G + Tobramycin	Biofilm-dispersing enzyme + Aminoglycoside antibiotic	*P. aeruginosa* (PAO1 and PAO1 P_BAD_psl ΔpelF	CV assay, MBEC assay (In-vitro) *C. elegans* (In-vivo)	Enhanced the antimicrobial efficacy of tobramycin by 10–100 folds and enhanced the *P. aeruginosa-*infected *C. elegans* survival.	[13]
Squalenyl Hydrogen Sulfate Nanoparticles	Alkylquinolone QS+ Tobramycin	QSI + Aminoglycoside antibiotic	*P. aeruginosa*	MBEC assay	Three-fold higher penetration and completely eradicated *P. aeruginosa* biofilms at almost eight times lower concentrations of tobramycin than the free drug and QSI alone	[203]
SLNPs	DNase I and levofloxacin	Enzyme + antibiotic	*P. aeruginosa* and *S. aureus*	Alamar-blue assay	SLNP formulation significantly increased the levofloxacin efficacy and markedly reduced the *S. aureus* and *P. aeruginosa* biofilm formation.	[204]
SLNPs	cis-2-decanoic acid (C2DA) + rifampicin	QSI inhibitor + antibiotic	*S. aureus* and *S. epidermidis*	CV assay	Demonstrated better anti-biofilm activity than free agents particularly in the biofilm formation stage, while unable to remove the preformed biofilms.	[205]
SLNPs	Anacardic acid + DNase	Antimicrobial agent + EPS-degrading enzyme	*S. aureus*	MBEC assay	Significantly reduced the MBIC and MBEC and markedly reduced the biofilm thickness and biomass demonstrated by CLSM	[206]
NLCs	DNase I and levofloxacin	Enzyme + antibiotic	*P. aeruginosa*	Alamar-blue assay	This formulation exhibited improved anti-biofilm activity against cystic fibrosis by decreasing the viscoelasticity in the patient’s lungs.	[204]
LNPs	*N*-acetyl-l-cysteine (NAC) + Moxifloxacin	Antimicrobial adjuvant + antibiotic	*S. epidermidis,* and *P. aeruginosa*	MBEC assay, CV assay, SEM biofilm analysis, MTT assay	NAC-loaded and unloaded moxifloxacin-LNPs significantly reduced the viable bacterial count, with no significant difference between the two. But NAC-loaded LNPs exhibited a safer profile compared to unloaded LNPs which is promising for in-vivo application.	[207]
Nanoemulsion	*Eucalyptus globulus* oil	Anti-biofilm agent	*P. aeruginosa* and *Candida* spp.	Calcofluor staining, atomic force microscopy.	Nanoencpsulated oil demonstrated improved anti-biofilm activity against *Candida* spp. (10-fold reduction in CFU) but was infective against *P. aeruginosa* biofilm due to less oil concentration (only 5%) being ineffective.	[208]
Nanosphere	Acylase + Gentamicin	QQ enzyme + Aminoglycoside antibiotics	*P. aeruginosa*	CV assay, fluorescent microscopy	Inhibit 95% of *P. aeruginosa* biofilm biomass production at 0.125 × 10^13^ NSs mL^−1^.	[209]

### 4.2. Lyotropic Liquid Crystal Nanoparticles

An evolving class of drug delivery systems is lyotropic lipid liquid crystalline nanoparticles (LCNPs). They resemble liposomes but are comprised of complex two-dimensional and three-dimensional non-lamellar nanostructures i.e., cubic mesophases and inverse hexagonal. Moreover, various mesophases consist of; lamellar, hexagonal (normal or inverse), and cubic (discontinuous or inverse bicontinuous) phases as depicted in Figure 7 [210]. Liquid crystals not only have ordered alignment and optical properties like solid crystals, but also have fluidity, viscosity, and surface tension like liquids, and are of two types i.e., lyotropic and thermotropic [211]. Thermotropic liquid crystals upon temperature change are found in high-melting point ionic molecules without adding any aqueous solvent. Low-melting point nonionic amphiphiles have been demonstrated to produce lyotropic LCNPs upon addition to water at concentrations above their critical micelle concentration [212].

Reverse hexagonal (H2) and inverse bicontinuous cubic (Q) phases are of specific interest for drug delivery, mainly for biomacromolecules, for their ability to encapsulate drugs with different molecular weights and hydrophilicity while maintaining their stability [211]. Phase Q also produces a viscous gel that can aid epithelial membrane absorption in topical applications such as wound dressings or nasal gels [214]. Mostly LCNPs are composed of biologically derived amphiphiles, comprising fatty acids and lipids that may enhance their biocompatibility, and biodegradability compared to other biomedical nanomaterials i.e., liposomes, metallic nanoparticles, quantum dots, carbon nanotubes, specific polymeric nanomaterials. Glycerol monooleate and phytantriol are frequently used lipids to produce liquid crystals [215]. These amphiphilic lipids affect the mesophase structure through their molecular shape and concentration. Additionally, external factors like pressure, temperature, and pH are also involved in determining the mesophase [211].

LCNPs are appropriate self-assembly entities due to the amphipathic nature of the lipid molecules for the encapsulation of both lipophilic and hydrophilic agents. Compared with liposomes, their two- and three-dimensional nanostructures and improved surface area are expected to enhance drug payload, protect against oxidation, hydrolysis, and enzymatic degradation, and control release [213,216]. The lipid bilayer presence can also provide a protective effect on incorporated drugs by inhibiting enzymatic degradation. LCNPs are proven for the stabilization of macromolecules i.e., lysozyme, insulin, and various antimicrobial peptides, and can also protect them from physical and chemical degradation [113]. The improved surface area of the LCNPs interfacial region imparts a strong ability to exploit the biofilm dispersing enzyme activity, thereby facilitating further advancement in preclinical and clinical studies. Moreover, the fabrication of LCNPs is comparatively simpler (readers interested in the technique of LCNP fabrication are recommended to read the study by Thorn et al. [217]) than several polymeric nanoparticles and dendrimers that usually require complex synthetic methods, or metal nanoparticles that require surface modification and drug conjugation [218]. LCNPs are hypothesized to have advantages over liposomes due to their greater membrane surface area facilitating improved lipid-to-protein loading ratio, and increased safety to sensitive enzymes [13]. Comparative studies for LCNPs and liposomes for vaccine, and antigen delivery exhibited that LCNPs are more effective, especially for transdermal immunization owing to increased permeability [219]. Previously our research group successfully developed LCNPs for the effective delivery of PslG and tobramycin against *P. aeruginosa* biofilms [13]. 

This study demonstrated that LCNP formulations were able to protect the Ps1G from proteolysis (enhanced stability), triggered and controlled its release, and significantly reduced the bacterial load compared to unformulated solutions of PslG and tobramycin in vitro. Moreover in vivo in the *C. elegans* infection model, LCNPs encapsulating PslG in combination with tobramycin improved antimicrobial activity by 10-fold compared to unformulated solutions of PslG and tobramycin (Figure 8) [13]. This necessitates the further development of LCNPs encapsulated with PslG and tobramycin combination for clinical translation.

### 4.3. Solid-Lipid Nanoparticles 

Solid lipid nanoparticles (SLNs) were first introduced in 1990 as nanoparticle drug delivery systems [220]. The main feature of SLNs is that they are comprised of lipids that stay solid at room temperature. They are usually composed of fatty acids, triglycerides, steroids, and waxes [221]. The average particle size of SLN is in the submicron range, ranging from approximately 40–1000 nm. The stability of the system is usually ensured by surfactants [222]. These lipid nanocarriers have the advantage of biocompatibility, sustained drug release, improved drug stability, targeted drug delivery, and a simple amplification process [221,223]. There has been limited research into the application of SLNs as carriers of anti-biofilm agents [223]. Krishna et al. developed SLNs encapsulating silver sulfadiazine and added them with DNase I to decrease the fibroblast toxicity and combat the biofilm-mediated resistance [161]. In vivo studies exhibited complete wound healing in 21 days after treatment with SLNPs encapsulating silver sulfadiazine and DNase I combination as compared to marketed drugs which displayed incomplete healing even after 21 days. This study suggests that this combination treatment is a promising therapeutic option for the treatment of biofilm-associated wound infections and accelerated wound healing [161].

DNase I and levofloxacin were encapsulated in SLNs for lung delivery and their antibacterial and anti-biofilm activities were evaluated against *S. aureus* and *P. aeruginosa* biofilms which exhibited excellent antimicrobial profiles [204]. In recent research, rifampicin and cis-2-decanoic acid (C2DA) were encapsulated into SLNs and tested for antibiofilm activity against staphylococcal biofilm during the formation and eradication phases [190]. The particle size and zeta potential were 127.2 ± 2.8 nm and 19.0 ± 7.64 mV respectively, with an encapsulation efficiency of approx. 69% for rifampicin and 46% for C2DA respectively. These SLNPs did not demonstrate any chemical interaction between the drugs and lipids shown by DSC studies and were stable for one year. Moreover, SLNs demonstrated better ant-biofilm activity than free agents against *S. epidermidis* and *S. aureus* in vitro particularly in the biofilm formation stage, though they were unable to remove preformed biofilms. These data suggest that a combined strategy of delivering C2DA and rifampicin via nanoparticle systems will pave the way for the development of strategies against biofilms [205]. A novel QSI, PqsR antagonist, was loaded into SLNPs to increase its mucus penetration and pulmonary delivery to effectively treat the *P. aeruginosa*-associated biofilm in cystic fibrosis patients [224]. The anti-virulence activity of nano-encapsulated QSI was seven times higher than that of the free compound. These startling results represent the novel perspective of extreme significance in the field of lipid-based nano-delivery of new anti-infective drugs.

Anjum et al. developed SLN loaded with anacardic acid through the homogenization method and further coated with DNase and chitosan [206]. Chitosan coating was done to yield cationic SLNs to enhance the attachment with *S. aureus* biofilm leading to increased eDNA degradation by DNase. The improved impact of SLN formulation on the minimum biofilm inhibition concentration (MBIC) and minimum biofilm eradication concentration (MBEC) compared to the control proved the formulation was a promising anti-biofilm strategy. SLN formulation significantly (*p* < 0.05) reduced the biofilm thickness and biomass shown by CLSM. Overall, the results demonstrated the improved efficacy of the developed formulation in overcoming biofilm-mediated antimicrobial resistance.

### 4.4. Nanostructured Lipid Carriers

Nanostructured lipid carriers (NLCs) have been extensively explored as lipid-based drug delivery systems. They hold a solid matrix at room temperature and are found to be better than SLNs due to higher drug-loading capacity (as drugs show greater solubility in oil compared to solid lipids) and lower water content of the particle suspension and avoid/minimize potential release of active compounds during storage [225]. However, NLCs suffer from the disadvantage of difficult surface functionalization [226]. To improve biocompatibility and excellent formulation properties, SLNs have been modified by substitution of solid lipids with liquid lipids to form NLCs. NLCs usually range in size from 100–500 nm and are produced using a mixture of solid lipids and liquid lipids i.e., oils, preferably in a ratio of 70:30 (ratios up to 99.9:0.1) with better bioactive retention and controlled release properties compared to SLNs. The total solid content of NLC can be increased to 95% [222]. NLCs are mostly composed of fatty acids, partial glycerides, triglycerides, steroids, and waxes. 

The hydrophobic core of NLCs and SLNs offers a suitable environment for the entrapment of hydrophobic drugs [227]. Since lipid nanoparticles are produced from physiological or biodegradable lipids, NLC shows good biocompatibility and tolerability. NLCs are divided into three types: one is to employ lipids with different structures to form NLCs; the other is to employ amorphous lipids to form NLCs; the third and most used NLC system consists of a mixture of liquid and solid lipids [170]. Like LCNPs and SLNPs, there are only limited studies available for the antibiotics and antimicrobial adjuvants co-delivery through encapsulation in NLCs. 

In a study, NLCs were developed for the co-delivery of levofloxacin and DNase against recalcitrant *P. aeruginosa* lung infection. The NLC formulation demonstrated a higher entrapment efficiency (Approximately. 60%) and biphasic drug release behavior over 2 days. Anti-biofilm activity was evaluated with the Live/Dead BacLight^®^ kit showing the untreated biofilm predominantly consists of live bacteria (green stained). However, after 30 min of exposure to the NLC formulation, red populations started to appear, suggesting the incidence of damaged bacteria and loss of membrane integrity. After one hour, only a few green bacteria were seen compared to red bacteria, indicating that the biofilm-embedded bacteria after treatment were severely damaged. After 24 h of NLC treatment, all bacteria were stained red due to the anti-biofilm activity of the levofloxacin and DNase-loaded nanoparticles [204].

### 4.5. Other Novel LNCs

A novel lipid nanocarrier was developed with a newly synthesized amphiphilic lipid squalene hydrogen sulfate for the co-encapsulation of hydrophilic tobramycin and a lipophilic alkyl quinolone QSI with a remarkably high loading capacity of 30% and approximately 10% respectively. This LBNC system produced a three-fold higher penetration and completely eradicated *P. aeruginosa* biofilms at almost eight times lower concentrations of tobramycin than the free drug and QSI alone [203].

## 5. Current Perspective and Future Directions

Biofilm-associated infections remain a critical healthcare problem worldwide, requiring the development of new and innovative strategies. Combination therapy using adjuvants with antibiotics proves to be a promising strategy, and our review provides the necessary information to advance this strategy and for the development of new therapeutics.

Adjuvant antimicrobial combination therapies have attracted substantial interest in the past decade and are currently emerging as a promising approach, with a special focus on restoration and the repurposing of different redundant antibiotics. Despite this remarkable preclinical success, the current list of approved antibiotic-adjuvant combinations comprises only one direct resistance disruptor, β-lactamase inhibitors. Other combinations of various antibiotics and adjuvants are yet to be approved. Some drug candidates, such as SPR741-a membrane-targeted adjuvant, are in clinical trials, and specific compounds, such as pentamidine analogs, are in preclinical development. Despite this, for the overall success of novel antimicrobial adjuvants, it is necessary to revisit current strategies from different perspectives.

There is substantial merit in the development of broad-spectrum antimicrobial adjuvants that can repurpose or improve several antibiotics against various bacterial pathogens. It can be accomplished by targeting the significant resistance mechanisms i.e., reduced permeability and antibiotic efflux. In this regard, we have discussed various potential EPS-degrading enzymes and QSI with no inherent antimicrobial activity and toxicity. Nontoxic and inactive adjuvants need to be explored as they warrant improved biocompatibility and reduce the chances of developing bacterial resistance. When structure-activity relationship (SAR) and basic chemistry benefit from finding better designs, these methods could be used in preclinical studies. Preclinical studies such as pharmacokinetics, pharmacodynamics, and effectiveness in various infection models, plasma protein binding, and in-depth toxicology studies are critical for these antimicrobial adjuvants. For physical combinations of antibiotics and antimicrobial adjuvants, it is important to conduct pharmacokinetic and biodistribution studies to confirm adequate bioavailability of the two distinct components at the site of infection site at various time points. 

Furthermore, comprehensive mechanistic studies are a prerequisite for various antimicrobial adjuvants like EPS-degrading enzymes and QSI to identify and exclude any off-target side effects that exist. Apart from this, the impact of antibiotics on adjuvants also required to be explored. Most studies used representative candidate antibiotics from various classes. Recent literature suggests that the enhancing activity of adjuvants is largely dependent on the combined antibiotic. This might be due to the better intrinsic interactions between antibiotics and bacteria or the synergistic interactions of certain antibiotics with adjuvants. These observations should be taken seriously and further explored. Different classes of antibiotics and several antibiotics from the same class must be studied for a range of adjuvants to comprehend the mechanism of action and the role of the physicochemical properties of antibiotics in synergistic combinations. This was recently examined with pentamidine-antibiotic combinations [228].

All the studies reviewed here emphasized single-species biofilms, though multi-species biofilms are equally big clinical and industrial problems [229,230]. Only a few studies have investigated the dispersion of multispecies biofilms [231,232,233,234] and none with a combination therapy of antibiotics and adjuvants. Given the heterogeneity of their composition, multispecies biofilms are often more difficult to eradicate than single-species biofilms, and studies on the molecular interactions among contributing species must elucidate potential therapeutic pathways (single or combination strategies) and develop the field of biofilm.

According to most of the studies discussed in this review, combined treatment with antibiotics and adjuvants was more effective for biofilm inhibition and eradication than antibiotic treatment alone. This trend was distinct across all classes of antibiotics, implying that a two-pronged strategy to disperse and eradicate biofilm could be translated into a successful therapeutic strategy. Additionally, simultaneous delivery or the release of two drugs at the site of the target appears to have clear advantages, an effect demonstrated by encapsulation into LNCs. However, few LNCs with combination therapy have been investigated for their potential against biofilms. This review highlights several effective antibiotic and adjuvant combination treatments that would be ideal to augment anti-biofilm potential following their encapsulation into LNCs.

Despite the advantages of certain combination therapies, most studies specifically targeting biofilms have been performed in vitro with non-clinically relevant models and therapeutic regimens. Few studies have made progress in vivo and even fewer have been evaluated in humans. Moreover, of the hundreds of newly discovered antimicrobial adjuvants mentioned in the literature, only limited have been investigated for in vitro activity, comparing single agents to combination treatments. It pinpoints that there is still substantial intact potential for the development of novel effective anti-biofilm combination treatments. Prospective studies of novel anti-biofilm drugs must follow early-test combination therapeutic designs, first investigating the drug’s in vitro antibiotic-enhancing activity, and their translation to in vivo models. Although this process may seem exhaustive, it identifies promising combinatorial lead compounds for clinical development. 

Antibiotics and adjuvant combinations should also be studied against metabolically inhibited bacterial subpopulations (e.g., stationary-phase or persister bacterial cells) which permits their application in real clinical settings. This also requires careful study of the exact mechanisms of this combination’s activity and the impact of adjuvants on bacterial virulence and QS. It has been stated previously that some adjuvants can have multiple functions in addition to their antibiotic synergistic effects. These compounds must be combined with antibiotics to examine their ameliorating efficacy in vitro, in vivo, and advanced infection models.

Overall, the potential for commercialization of combination therapies is tremendous. A research ecosystem is thriving and new ways to repurpose existing antibiotics are being discovered. However, efforts are needed to advance such compounds to preclinical and clinical development stages. There are many promising agents in the literature that could be employed as antimicrobial adjuvants. An enhanced understanding of the various aspects of adjuvant therapy at the design, genetic, biochemical, and preclinical stages can certainly provide new dimensions to this upcoming field and guarantee their future success. To further facilitate this, combination treatment through encapsulation in nanoparticulate systems is investigated against biofilm. Lipid nanocarriers have proven to be a promising alternative for antimicrobial delivery owing to their superior properties compared to conventional formulations in the market, owing to their biocompatibility, improved drug loading capacity, protection from chemical or enzymatic degradation, controlled drug release, higher bioavailability, targeted delivery, ease of preparation and scale-up feasibility. Furthermore, due to the compositional versatility and surface modification of LNCs, products can be designed with specific physicochemical properties [235]. LNCs encapsulating a combination of antibiotics and antimicrobial adjuvants exhibited enhanced or retained antimicrobial activity compared to free drug or unformulated solutions in both in-vitro and in-vivo experiments against biofilms. Nonetheless, studies demonstrated the significance of adopting a rational approach in the development of antimicrobials encapsulated LNCs, yet careful selection of components is critical to its efficacy [173]. Another interesting LNC i.e., lipid-polymer hybrid nanoparticles, given its several beneficial attributes in successful anti-cancer and other active targeted drug deliveries [236], is yet to be explored for the co-delivery of antibiotics and adjuvants. 

Furthermore, limitations in the current development of antimicrobials encapsulated LNCs include penetration, achieving biofilm targeting to blood circulation, and accumulation across the entire thickness of the infectious biofilm, accompanied by deep killing within the biofilm. To prevent the recurrence of infection, complete killing is crucial, which is a difficult problem in clinical infection treatment. Moreover, in vitro methods require standardized procedures to ensure their homogeneity and reduce variability in results [237]. In addition, only limited in vivo studies have been carried out for the co-encapsulation of antibiotic and antimicrobial adjuvant combinations in LNCs for the treatment of biofilms. Therefore, advancing proof-of-concept for the efficacy of antimicrobials-loaded LNCs is critical for realizing their commercial potential. Antimicrobial-loaded LNCs can progress current clinical drug treatments, provide innovative products, and save discarded antibiotics. Additionally, the rising threat of drug-resistant strains of bacteria could be minimized, thereby diminishing the scarcity of new antibiotics. Hence, LNCs encapsulating antimicrobials open a new horizon for saving millions of lives and preventing the catastrophic effects of bacterial infections.

Without an appropriate evaluation of the multifactorial parameters recognized to affect antibiofilm therapies, realizing the clinical benefits offered by these therapies and delivery systems will remain a challenge. It will take a collaborative effort of microbiologists, chemists, engineers, and medical professionals, combined with comprehensive mechanistic, pharmacokinetic, pharmacodynamic, and bactericidal studies, to accurately evaluate the efficacy of these promising advanced strategies for clinical translation. However, the adoption of these new therapies will also require significant advancement in the diagnosis of biofilm infections, regulatory classification of clinically feasible treatments, and collaborations between regulators and industry partners to deliver anti-biofilm treatments to patients.

## Figures and Tables

**Figure 1 pharmaceutics-16-00396-f001:**
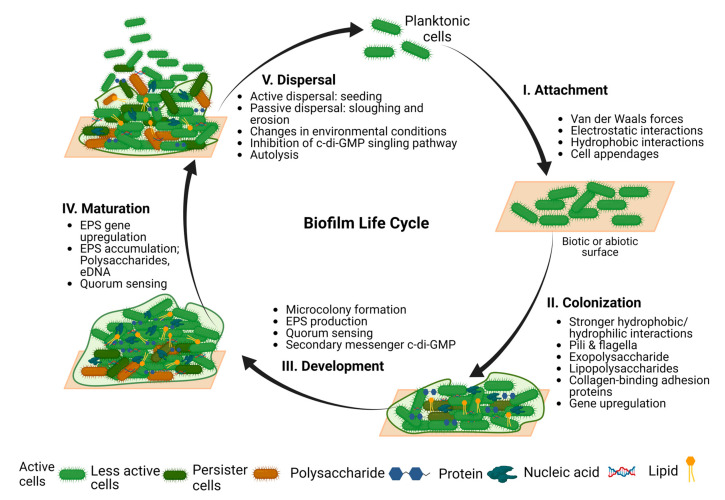
The biofilm formation cycle consists of five main stages: I, attachment; II, colonization; III, development; IV, maturation; V, active dispersal. This figure was created with BioRender.com.

**Figure 2 pharmaceutics-16-00396-f002:**
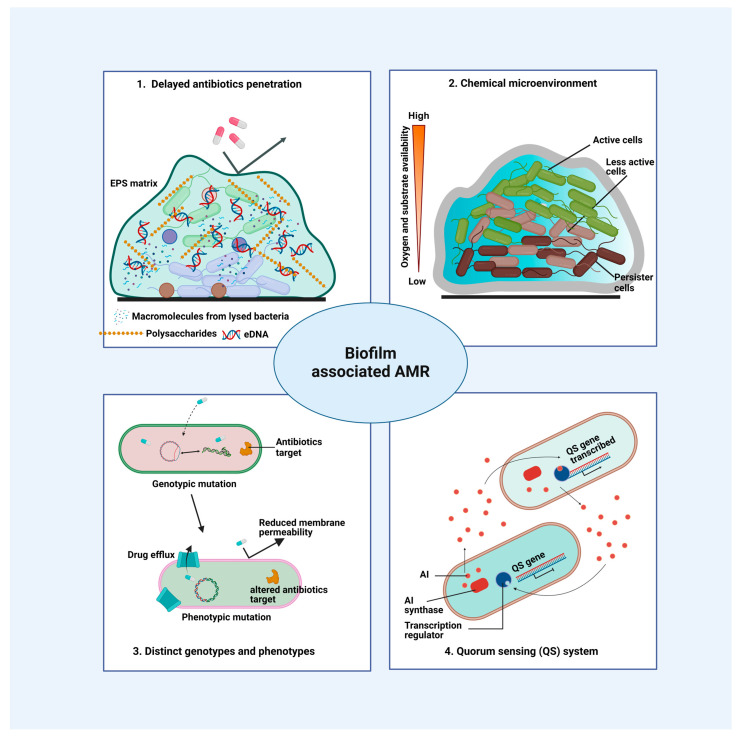
An illustration of the different mechanisms of biofilm-mediated resistance to antimicrobials. The figure was created with BioRender.com.

**Figure 3 pharmaceutics-16-00396-f003:**
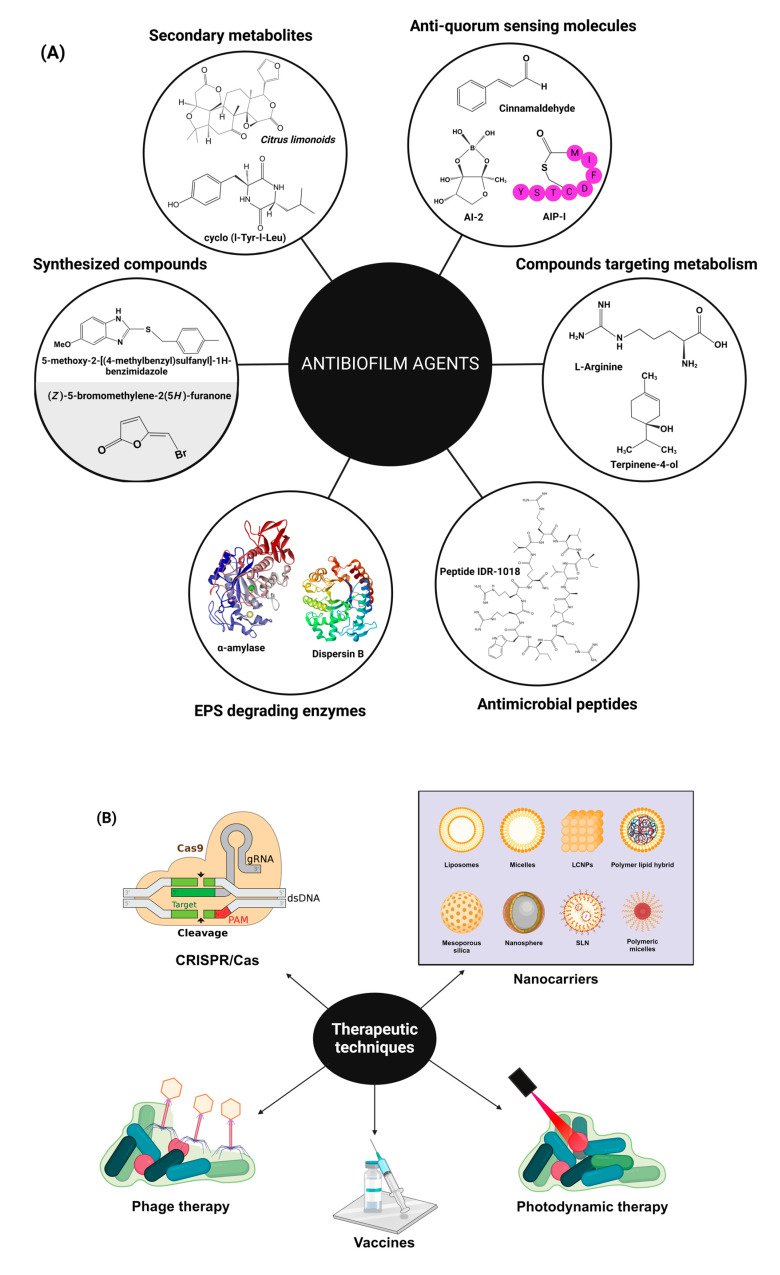
(**A**). an illustration of the various antibiofilm agents that target the different compounds that are responsible for biofilm formation. (**B**). an illustration of the various therapeutic techniques involved in directly targeting the biofilm formation process. This figure was created with Biorender.com.

**Figure 4 pharmaceutics-16-00396-f004:**
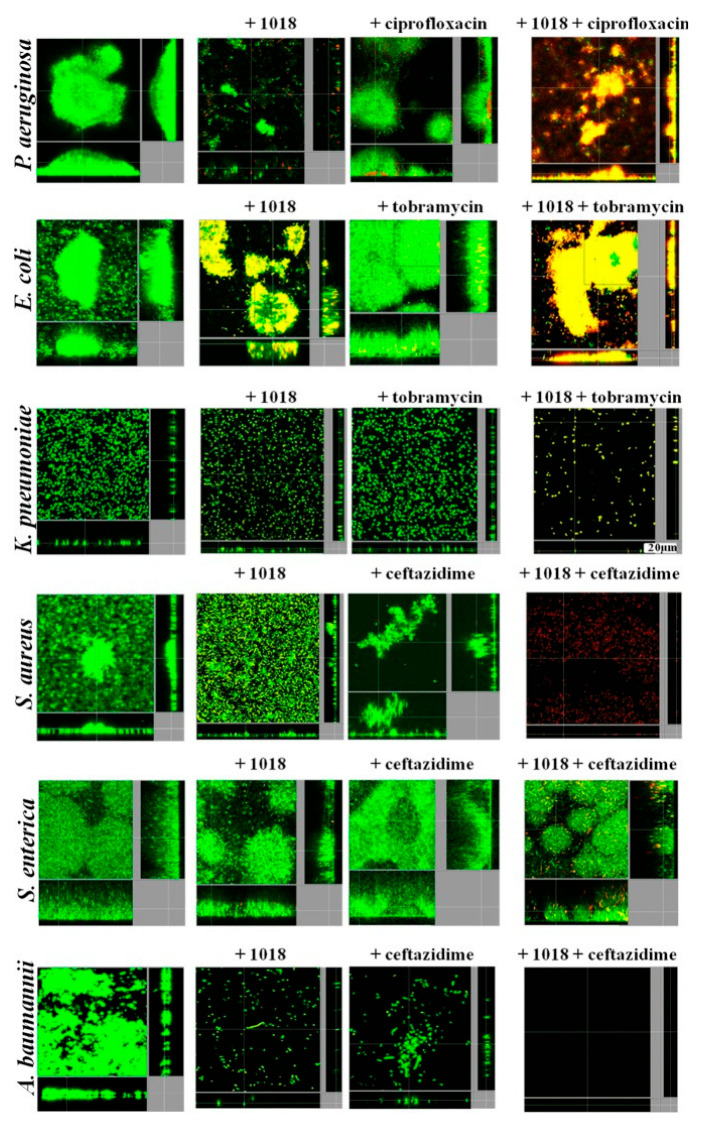
Synergy of antibiofilm peptide 1018 with conventional antibiotics in eradicating preformed biofilms. Biofilms were grown in a flow cell. Treatments (antibiotics, peptide, or combinations) were added after 2 days of biofilm growth and continued for 24 h. After 3 days, before confocal imaging, the bacteria were stained green with the all-bacteria stain Syto-9 and red with the dead-bacteria stain propidium iodide (merged images shown as color change from yellow to red). Each panel shows reconstructions from the top of the large panel and the sides of the right and bottom panels (x-y, y-z, and x-z dimensions, respectively). This figure was reproduced from Reffuveille et al. [63]. Copyright 2014, with permission from the American Society for Microbiology.

**Figure 5 pharmaceutics-16-00396-f005:**
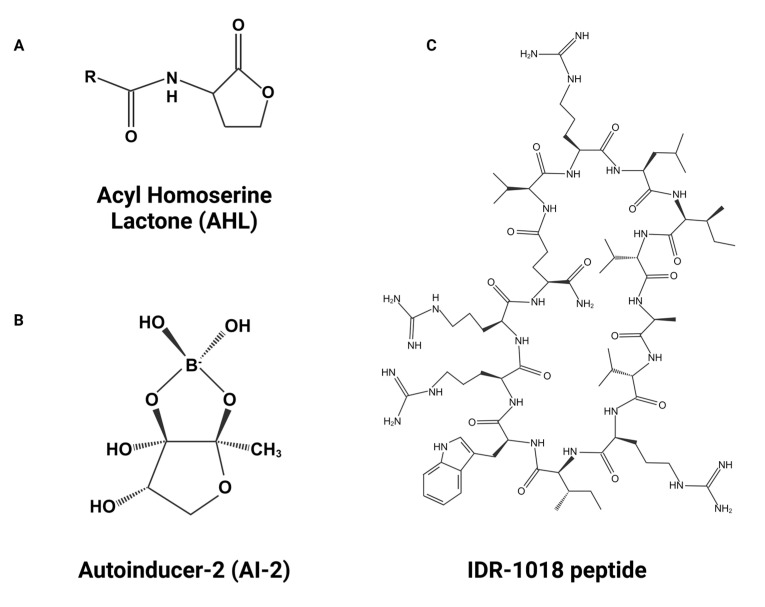
The structures of quorum sensing signaling molecules: (**A**) acyl-homoserine lactone (AHL), (**B**) autoinducer-2 (AI-2), and (**C**) IDR-1018 anti-biofilm peptide.

**Figure 6 pharmaceutics-16-00396-f006:**
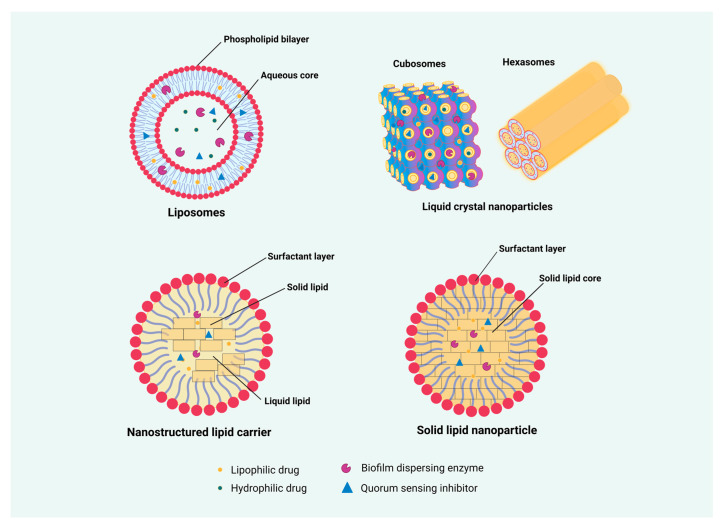
Various lipid nanocarriers for the co-delivery of antibiotics and antimicrobial adjuvants i.e., biofilm dispersing enzymes, and/or quorum sensing inhibitors to combat different bacterial biofilms. This figure was created with Biorender.com.

**Figure 7 pharmaceutics-16-00396-f007:**
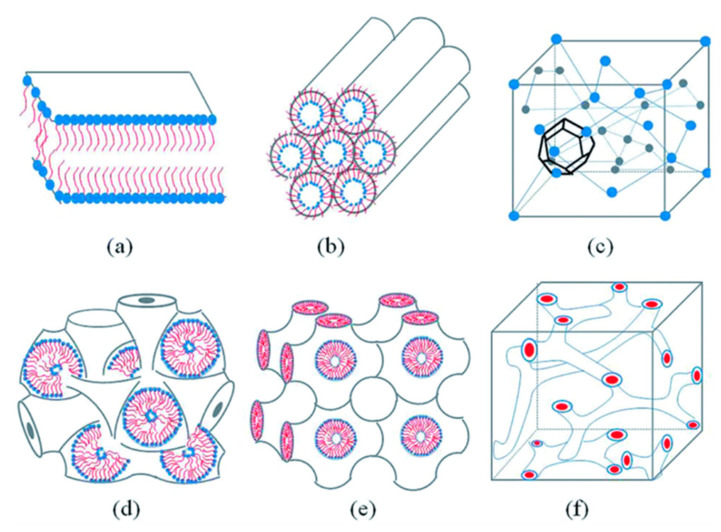
A schematic illustration of lyotropic liquid crystalline phases (**a**) Lamellar, (**b**) reverse hexagonal, (**c**) inverse micellar cubic (Fd3m), (**d**) inverse bicontinuous cubic phase (lm3m), (**e**), inverse bicontinuous cubic phase (Pn3m), and (**f**) inverse bicontinuous cubic phase (la3d). This figure is reproduced from Huang et al. [213] Copyright 2018, with permission from the Royal Society of Chemistry.

**Figure 8 pharmaceutics-16-00396-f008:**
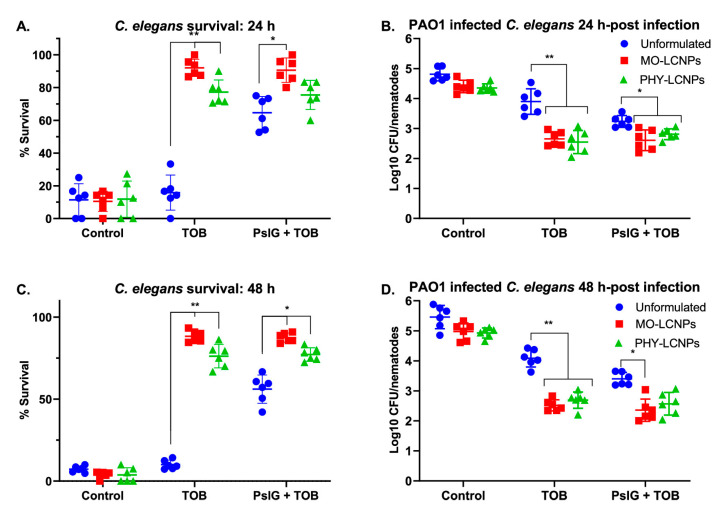
L4 stage *C. elegans* infected with PAO1 for 6 h following treatment with tobramycin (3 μg/mL) alone or in combination with PslG (20 nM)-as unformulated solution or encapsulated in MO-LCNPs (0.05 mg/mL MO) or PHY-LCNPs (0.05 mg/mL PHY). Nematode survival and bacterial burden (CFU) 24 h after infection was established (**A**,**B**), and Nematode survival and bacterial burden (CFU) 48 h after infection was established (**C**,**D**). Data are expressed as mean ± SD, n = 6, two-way ANOVA followed by Tukey’s multiple comparisons test, ** *p* < 0.01 and * *p* < 0.05. It is reproduced from Thorn et al. [13] Copyright 2021, with permission from the American Chemical Society.

**Table 1 pharmaceutics-16-00396-t001:** Potential quorum sensing inhibitors (QSI) and quorum quenchers (QQ) and their application against various biofilms.

Quorum Sensing Inhibitors/Quorum Quenchers	Targeted Component	Tested Organism	Result	Ref.
3-amino-7-chloro-2-nonylquinazolin-4(3H)-one (ACNQ)	Effective inhibitor of the PqsR receptor	*P. aeruginosa*	Complete eradication of 24 h *P. aeruginosa* biofilm infections.	[61]
(*5Z*)-4-bromo-5-(bromomethylene)-3-butyl-2(5H)-furanone (furanone C-30)	*afeI* and *afeR* genes in *Acidithiobacillus ferrooxidans*	*Acidithiobacillus ferrooxidans*	Successfully inhibited the EPS production and hence biofilm formation and significantly downregulated the gene expression involved in biofilm growth.	[90]
Trans-cinnamaldehyde	*E. faecalis* biofilm cells	*E. faecalis* biofilm	No significant increase in *E. faecalis* biofilm metabolic activity and no significant reduction in cell viability after long-term treatment.	[91]
Trans-cinnamaldehyde	*S. mutans* UA159 genes	*S. mutans*	Significantly reduced plaque formation in a rat carrier model.	[92]
AiiA lactonase enzyme (produced by engineered T7 bacteriophage)	3-oxo-C8-HSL (AHL) 3-oxo-C12-HSL (AHL)	*A. tumefaciens* *P. aeruginosa*	Degradation of 3-oxo-C_8_-HSL produced by *A. tumefaciens* KYC6 and degradation of 3-oxo-C_12_-HSL produced by *P. aeruginosa.*	[93]
*Β*-Sitosterol glucoside	AL-3	*E. coli*	Complete inhibition of *E. coli* O157:H7 motility ≥2-fold reduction of *E. coli* biofilm formation.	[94]
Quercetin	*P. aeruginosa* PAO1 Las and Rhl QS circuits	*P. aeruginosa*	Significantly reduced PAO1 biofilm formation (50% reduction) and inhibited PAO1 adhesion. Significantly reduced PAO1 QS gene expression (*lasI, lasR, rhlI, rhlR)*	[95]
Echinatin	AI-2	*E. coli* clinical isolated strains	Reduced biofilm EPS production and virulence factors.	[96]
Ajoene	Gac/Rsm QS circuit	*P. aeruginosa* and *S. aureus*	Reduced biofilm mass and sRNA expression.	[77]
Honey	*P. aeruginosa* biofilm	*P. aeruginosa*	Effective inhibition and eradication of *P. aeruginosa* biofilm. Significant reduction in living bacteria cells in *P. aeruginosa* biofilms at a sub-inhibitory concentration.	[97]
Naringenin	AHL uptake pathway	*E. coli*	Significant reduction in biofilm formation compared to un-formulated solution.	[98]
Zinc oxide nano spikes	*N*-acyl-homoserine lactone *P. aeruginosa las and psq* QS circuits	*P. aeruginosa*	Significant reduction in the production of virulent factors, and inhibited up to 80% of biofilm formation at half MIC.	[99]
Silver nanoparticles	*P. aeruginosa las, rhl and psq QS circuits*	*P. aeruginosa*	Reduction in PAO1 virulence gene expression, swarming activity, and biofilm formation.	[100]
Salicylic acid	*P. aeruginosa las* QS circuit	*P. aeruginosa*	Significant biofilm inhibition after 48 h.	[101]
2-methoxy-4-vinyl phenol (2M4VP)	LuxO active site	*V. cholerae*	Inhibited up to 50% of biofilm formation and repression of virulence genes.	[102]
Linear copolymers (*pMAA_25_-co-pMMA_75_* and *pIA_25_-co-pMMA_75_*)	3-oxo-C_6_-HSL, C_4_-HSL, C_6_-HSL	*V. fisheri*, *A. hydrophilia*	Reduced *V. fisheri* bioluminescence and *A. hydrophilia* biofilm formation.	[103]
(z)-5-octylidenethiazolidine-2,4-dione (TZD-C8)	LasI	*P. aeruginosa*	70% biofilm biomass reduction at MIC.	[104]
Cis-14-methylpantane-2-enoic acid	RpfFBc	*Burkholderia sp.*	Significant inhibition of biofilm formation and decreased QS-mediated virulence factors.	[105]
Palmitoleic acid Myristoleic acid	*abaR*	*A. baumannii* reference and clinical strains	Dispersed 24 h biofilm and inhibited 40% of biofilm formation.	[106]
CDC PF	*LasR transcription regulator*	*P. aeruginosa*	Formulated QSI prolonged biofilm treatment for 48 h, resulting in ≥20% inhibition.	[107]
*L. speciosa* extracts	QS genes	Sinusitis bacteria isolates	Significant antibiofilm activity, ≥ 50% biofilm inhibition, but ineffective against *S. aureus*.	[108]
Tea polyphenols (TP)	QS virulence	*K. pneumonia* and *C. violaceum*	Inhibited 23.7% of biofilm formation at half-MIC of TP, reduced *C. elegans* death (26.7%) at half-MIC of TP.	[109]

**Table 2 pharmaceutics-16-00396-t002:** Potential EPS-degrading enzymes and their application against various biofilms.

EPS-Degrading Enzyme	Targeted Component	Carrier System with/without Combined Antimicrobial	Test Organism	Result	Ref.
DspB	GlcNAc-(β-1,6)- GlcNAc	DspB loaded on Carboxymethyl chitosan nanoparticles	*A. actinomycetemcomitan, Staphylococcus aureus* and Staphylococcus *epidermidis*	Improved enzyme reusability and stability as well as enhanced biofilm inhibition and eradication efficacy compared to non-formulated solution.	[112]
DspB	-	Silver nanoparticles fused with DspB	*Staphylococcus epidermidis*	Enhance the biofilm eradication potential by 2-fold.	[113]
DspB	-	Fusion of DspB with magnetoreceptor protein and subsequently loaded onto Fe3O4@SiO2 nanoparticles	*Staphylococcus* sp., *Staphylococcus aureus, Pseudomonas Putida, Bacillus Cereus*	Enzymatic killing by DspB was increased with 40–60% biofilm removal.	[114]
DspB	-	DspB + triclosan coated on vascular catheters	*Staphylococcus aureus*	Enhanced the biofilm eradication biofilms compared with control, DspB alone, or triclosan alone, thus demonstrating synergistic anti-biofilm activity of the combination treatment.	[115]
DspB	-	DspB loaded onto Polyhydroxyalkanoate asymmetrical membranes	*Staphylococcus epidermidis*	Weakly inhibited the biofilm formation but effectively disrupted the preformed biofilms.	[116]
DspB DNase I	-	DspB + DNase I + Tobramycin	*S. aureus*	Combined treatment of tobramycin with either DspB or DNase I decreased bacterial load in *S. aureus* biofilms by 7500-fold and 8780-fold respectively, while tobramycin alone reduced cell numbers by only 40-fold. Combined treatment with both enzymes did not significantly enhance the tobramycin efficacy.	[117]
DspB	-	DspB + KSL-W peptide + Pluronic F-127	*MRSA, Vancomycin-resistant Enterococci, S. epidermidis, CoNS, A. baumannii, P. aeruginosa, K. pneumoniae*	More effective in biofilm-killing than the extensively used commercial silver-based antimicrobial product Silver-Sept^®^.	[118]
DspB	-	DspB wound spray combined with Acticoat™,	*MRSA, S. epidermidis*, *A. baumannii*, and *K. pneumoniae*	Combined with treatment with antimicrobial silver wound dressing Acticoat™, the spray decreased viable cell count by 80% compared to a 14% decline with wound dressing alone.	[119]
PgaB	GlcN-(β-1,6)-GlcN	Unformulated solution + gentamicin	*E. coli*, *S. carnosus, S. epidermidis*, *Bordetella pertussis*	Potentiate the efficacy of gentamicin in biofilm killing.	[120]
PslG	Man*p*-(β-1,3)- Man*p*	Unformulated solution and immobilised on material surfaces	*P. aeruginosa*	Covalently bound PslG significantly reduced *P. aeruginosa* biofilm formation and surface attachment by ~99.9% (~3-log) compared to untreated surfaces.	[121]
Ps1G	-	Immobilization of Ps1G on medical-grade commercial catheter tubing	*P. aeruginosa*	3-log and 2-log reduction in bacterial load after 11 and 14 days of enzyme post immobilization respectively. In vivo showed ~a 1.5-log reduction after 24 h) in the colonization of the clinical *P. aeruginosa* strain.	[122]
Ps1G Pe1A	-	Unformulated solutions + Colistin	*P. aeruginosa*	Effectively inhibited the biofilm formation and significantly disrupted the preformed biofilm with a 58–94% reduction in biofilm biomass. Pe1A also potentiated the colistin efficacy with approx. 50% neutrophil killing.	[123]
PelA, Sph3	(1,4)	Unformulated solutions + amphotericin B, caspofungin, and posaconazole,	*A. fumigatus*	Effectively disrupted the biofilm with an EC50 of approx. 0.4 nM for both enzymes. PelA and Sph3 also enhanced the antifungal efficacy through increased intracellular penetration.	[124]
Alginate lyase Ps1G	Alginate, Man*p*-(β-1,3)- Man*p*	Unformulated solutions	Mucoid *P. aeruginosa* (Clinical isolate)	The comparative study with both glycoside hydrolases showed significant biofilm formation inhibition.	[125]
Alginate lyase	Alginate	Unformulated solution + ciprofloxacin and tobramycin	*P. aeruginosa*	AL enhanced the efficacy of antibiotics through biofilm disruption leading to the significant reduction of biofilm biomass.	[126]
Alginate lyase	-	Unformulated solution +Gentamicin and Ceftazidime	*P. aeruginosa*	The synergy of AL and gentamicin significantly eliminated the mucoid bacteria from biofilm while ceftazidime with AL was more effective against non-mucoid strains.	[127]
Alginate lyase	-	AL immobilized on CS nanoparticles in combination with ciprofloxacin	Mucoid *P. aeruginosa*	Developed nanoparticles significantly inhibited the biofilm formation, and reduced biofilm biomass, density, and thickness in preformed biofilm.	[128]
Alginate lyase	-	AL and levofloxacin in high methoxylated pectin microsphere hydrogel	*P. aeruginosa*	AL enhanced the antimicrobial efficacy of levofloxacin by 35% compared to unformulated solutions.	[129]
Alginate lyase	-	AL immobilized on bacterial cellulose membranes + Gentamicin	*P. aeruginosa*	The combination therapy exhibited a synergistic effect resulting in an 86.5% reduction in viable bacterial cells.	[130]
DNase I	eDNA	Unformulated solution + ampicillin, cefotaxime, rifampicin, levofloxacin, and azithromycin	*Escherichia coli,* *Haemophilus influenzae, Klebsiella pneumoniae, Pseudomonas aeruginosa, Staphylococcus aureus, Streptococcus pyogenes, and Acinetobacter baumannii*	Combined treatment with DNase I enhanced antibiotic efficacy resulting in reduced biofilm biomass and CFU count.	[131]
DNase I	-	Unformulated solution + Mg^2+^	*P. aeruginosa* alone or mixed species biofilm with *Enterococcus faecalis, Salmonella Typhimurium, and S. aureus*	Combined treatment of DNase I with Mg^2+^ caused 90% biofilm reduction within 5 min against preformed *P. aeruginosa* biofilms. While this combination was less effective in treating biofilms of mixed species.	[132]
DNase I	-	DNase I + Ceftazidime or + ceftazidime linked with chitosan	*Burkholderia pseudomallei*	DNase I + Ceftazidime caused a 3–4 log reduction in viable cell count in a 2-day-old biofilm. While DNase I + ceftazidime linked with chitosan also significantly inhibited and eradicated pr-formed biofilm.	[133]
DNase I	-	DNase-loaded-polylactic-glycolic acid (PLGA) nanoparticles	*S. aureus* and *P. aeruginosa*	This combination was effective in preventing biofilm formation and removed >99.8% of the established biofilms.	[134]
DNase I	-	Polymer-encapsulated DNase I (n(DNase))	*S. aureus*	Effective penetration and log retention time of n (DNase) in biofilm compared to DNase I alone led to 92.2% biofilm disintegration.	[135]
DNase I	-	Chitosan nanoparticles (CS NP) loaded with ciprofloxacin and functionalized with DNase I	*P. aeruginosa*	DNase-functionalized NPs demonstrated significant biofilm formation inhibition and a 2.5-fold reduction in biofilm biomass in preformed biofilm compared to unformulated solutions.	[136]
DNase + Proteinase K	eDNA Exoproteins	Unformulated solutions	Multispecies oral biofilms	DNase I significantly inhibited the growth of *Fusobacterium nucleatum*, *Actinomyces oris*, Streptococcus *oralis Streptococcus mutans*, and *Candida albicans.* Proteinase K caused a significant increase in *S. oralis* and *S. mutans* CFUs but reduced the *V. dispar* and *C. albicans* CFUs compared to control. CLSM results showed significant biofilm disruption with combined treatment.	[137]
α-amylase + Pancreatic protease type-1(PtI)	Exoproteins	Unformulated solutions	*S. aureus*, *MRSA*, *E. coli*	The enzyme combination exhibited significant inhibition of established biofilm (90%, 93%, and 78%), and biofilm prevention (51%, 70%, and 44%) against *S. aureus*, *MRSA*, *and E. coli* respectively.	[138]
Serine endo-peptidase protease (Alcalase 2.4 L FG)	-	Oxacillin and penicillin G encapsulated in shellac nanoparticles followed by coating with Alcalase 2.4 L FG	*S. aureus*	Enhanced the efficacy of antibiotics (~1 × 10^6^ CFU/mL reduction as compared to antibiotic alone) and also exhibited a prompt biofilm degradation.	[139]
Alcalase 2.4 L FG	-	Protease functionalized Carbopol nanogels	*S. aureus*, *S. epidermidis*, *P. aeruginosa*, *K. pneumoniae*, *E. faecalis*, and *E. coli*	Nanogels caused a 6-fold reduction in biofilm biomass and a significant decrease in cell density compared to unformulated solutions. Co-treatment of ciprofloxacin and Alcalase coated nanogels produced a 3-log decrease in viable cell count which further led to an undetectable number following co-encapsulation of ciprofloxacin and alcalase in nanogel.	[140]

## Data Availability

Not applicable.

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
