# Peer review of "Lipid Nanocarriers-Enabled Delivery of Antibiotics and Antimicrobial Adjuvants to Overcome Bacterial Biofilms"

_pharmaceutics, 2024, doi:10.3390/pharmaceutics16030396_

Round 1

Reviewer 1 Report

Comments and Suggestions for Authors

The manuscript of Prestidge and collaborators is an excellent review on the use of lipid nanocarriers to tackle the problem of biofilms and bacterial resistance to most antibiotics. Publication is warranted but some additional requests and remarks were formulated to further improve the quality of the paper, as well as some linguistic suggestions to improve readability.

Detailed stepwise remarks:

line 57:   regarding the economic burden, insert for clarity “based on a 2014 study”. In addition, is a more recent number maybe available?

line 74: the authors write "and also due", but the following sentence is however not a reason for low concentrations. Correct to "and in addition there is a reduced availability ..." as these are separate statements.

Line 80: delete the word “of”.

line 174: add the word "of" in “presence of persister cells”.

Figure 2: the abbreviation AI (autoinducers) is not explained (at its first use). A sentence could be added maybe to this paragraph at lines 221-234 to better situate the term.

line 308: include: "(IDR-1018; the innate defense regulator peptide-1018 is a 12-mer cationic peptide)”

line 312-314 describe the effects of combined use of ciprofloxacin and peptide 1018. However, figure 4 only shows resistance against ciprofloxacin. It seems at least one panel in the figure is missing then as the synergistic effect of  the combination is not shown?

line 330: repurposing ... This sentence is not adequately explained. The clue message of reference 65 should be added. Why repurposing is needed to develop adjuvants? Why not with existing antibiotics?

Page 10, Quorum sensing inhibitors, first paragraph:  both, AHL and AI-2 structure could be added in a figure - maybe also include then the peptide 1018 structure.

line 534: after "domain" insert  "for biofilm disruption". The enzyme does not hold antibacterial activity by itself.

line 594: insert "which" in combination with ..... In general, too long sentences hamper fluid reading.

line 665: the term “LNCs” was already explained in previous paragraph. Unnecessary to repeat.

line 672: in contrast, LCNPs is not yet explained, except in the abstract, and further down at line 756. It is better to do this here at first use. Likewise, the authors should already point to section 4.2 as well, for a discussion on what the LCNPs exactly are.

Figure 5: the figure does not really show a difference between liposomes and LCNP and therefore is somewhat awkward. Some justifying text should be included either in the figure or its legend.

line 775: P. aeruginosa should be in italics.

line 813-814: incomplete sentence; please correct.

At some places, like at line 852, it has been stated that the fabrication of LCNPs is simpler than several polymeric nanoparticles. I believe this is an important message, but unfamiliar to most readers. Can the authors briefly document/describe this technique, or at least point the reader to the best possible references on the fabrication of these particles?

Finally, but important, in view of the efforts of all authors, I would like to invite the senior author to reflect on the order of the author list. 

Comments on the Quality of English Language

see report 

Reviewer 2 Report

Comments and Suggestions for Authors

In this review, the authors summarized “Lipid Nanocarriers Enabled Delivery of Antibiotics and Antimicrobial Adjuvants to Overcome Bacterial Biofilms”. However, further clarification is required to recommend the review articles for publication.

Some critical comments and suggestions

1.      The graphical abstract lacks scientific detail and should be revised to better reflect the content.

2.      Figure 1 is overly generalized; it should focus more on molecular mechanisms of biofilm formation.

3.      In Figure 3, the distinction between different types of antibiofilm agents is unclear. For instance, differentiate between natural compounds, secondary metabolites, and compounds targeting cellular components.

4.      Figure 3 lists organic and inorganic nanoparticles under therapeutic techniques without clear categorization. Clarify how these fit into the classification.

5.      The stability of antimicrobials in lipid-based nanocarriers needs explanation, along with factors influencing their formation.

6.      Table 1 includes content beyond lipid nanocarriers, deviating from the article's aim. Content should align closely with the article's focus.

7.      The composition of lipid nanocarriers and the omission of antibiotics require clarification from the authors.

Comments on the Quality of English Language

Minor editing of English language required

Round 2

Reviewer 2 Report

Comments and Suggestions for Authors

-